# Efficient and accurate extraction of in vivo calcium signals from microendoscopic video data

Pengcheng Zhou[1,2,3,4,5]*, Shanna L Resendez[6†], Jose Rodriguez-Romaguera[6†], Jessica C Jimenez[7,8,9†], Shay Q Neufeld[10], Andrea Giovannucci[11], Johannes Friedrich[11], Eftychios A Pnevmatikakis[11], Garret D Stuber[6,12,13], Rene Hen[7,8,9], Mazen A Kheirbek[14,15,16,17], Bernardo L Sabatini[10], Robert E Kass[1,3,18], Liam Paninski[2,4,5,7,19,20]

[1]Center for the Neural Basis of Cognition, Carnegie Mellon University, Pittsburgh, United States; [2]Department of Statistics, Columbia University, New York, United States; [3]Machine Learning Department, Carnegie Mellon University, Pittsburgh, United States; [4]Grossman Center for the Statistics of Mind, Columbia University, New York, United States; [5]Center for Theoretical Neuroscience, Columbia University, New York, United States; [6]Department of Psychiatry, University of North Carolina at Chapel Hill, Chapel Hill, United States; [7]Department of Neuroscience, Columbia University, New York, United States; [8]Division of Integrative Neuroscience, Department of Psychiatry, New York State Psychiatric Institute, New York, United States; [9]Department of Psychiatry & Pharmacology, Columbia University, New York, United States; [10]Department of Neurobiology, Harvard Medical School, Howard Hughes Medical Institute, Boston, United States; [11]Center for Computational Biology, Flatiron Institute, Simons Foundation, New York, United States; [12]Department of Cell Biology and Physiology, University of North Carolina at Chapel Hill, Chapel Hill, United States; [13]Neuroscience Center, University of North Carolina at Chapel Hill, Chapel Hill, United States; [14]Weill Institute for Neurosciences, University of California, San Francisco, San Francisco, United States; [15]Neuroscience Graduate Program, University of California, San Francisco, United States; [16]Kavli Institute for Fundamental Neuroscience, University of California, San Francisco, San Francisco, United States; [17]Department of Psychiatry, University of California, San Francisco, San Francisco, United States; [18]Department of Statistics, Carnegie Mellon University, Pittsburgh, United States; [19]Kavli Institute for Brain Science, Columbia University, New York, United States; [20]Neurotechnology Center, Columbia University, New York, United States

*For correspondence:
zhoupc1988@gmail.com

†These authors contributed equally to this work

Competing interests: The authors declare that no competing interests exist.

**Abstract** In vivo calcium imaging through microendoscopic lenses enables imaging of previously inaccessible neuronal populations deep within the brains of freely moving animals. However, it is computationally challenging to extract single-neuronal activity from microendoscopic data, because of the very large background fluctuations and high spatial overlaps intrinsic to this recording modality. Here, we describe a new constrained matrix factorization approach to accurately separate the background and then demix and denoise the neuronal signals of interest. We compared the proposed method against previous independent components analysis and constrained nonnegative matrix factorization approaches. On both simulated and experimental data recorded from mice, our method substantially improved the quality of extracted cellular signals and detected more well-isolated neural signals, especially in noisy data regimes. These

advances can in turn significantly enhance the statistical power of downstream analyses, and ultimately improve scientific conclusions derived from microendoscopic data.

DOI: https://doi.org/10.7554/eLife.28728.001

## Introduction

Monitoring the activity of large-scale neuronal ensembles during complex behavioral states is fundamental to neuroscience research. Continued advances in optical imaging technology are greatly expanding the size and depth of neuronal populations that can be visualized. Specifically, in vivo calcium imaging through microendoscopic lenses and the development of miniaturized microscopes have enabled deep brain imaging of previously inaccessible neuronal populations of freely moving mice (*Flusberg et al., 2008*; *Ghosh et al., 2011*; *Ziv and Ghosh, 2015*). This technique has been widely used to study the neural circuits in cortical, subcortical, and deep brain areas, such as hippocampus (*Cai et al., 2016*; *Ziv et al., 2013*; *Jimenez et al., 2018*; *Rubin et al., 2015*), entorhinal cortex (*Kitamura et al., 2015*; *Sun et al., 2015*), hypothalamus (*Jennings et al., 2015*), prefrontal cortex (PFC) (*Pinto and Dan, 2015*), premotor cortex (*Markowitz et al., 2015*), dorsal pons (*Cox et al., 2016*), basal forebrain (*Harrison et al., 2016*), striatum (*Barbera et al., 2016*; *Carvalho Poyraz et al., 2016*; *Klaus et al., 2017*), amygdala (*Yu et al., 2017*), and other brain regions.

Although microendoscopy has potential applications across numerous neuroscience fields (*Ziv and Ghosh, 2015*), methods for extracting cellular signals from this data are currently limited and suboptimal. Most existing methods are specialized for two-photon or light-sheet microscopy. However, these methods are not suitable for analyzing single-photon microendoscopic data because of its distinct features: specifically, this data typically displays large, blurry background fluctuations due to fluorescence contributions from neurons outside the focal plane. In *Figure 1*, we use a typical microendoscopic dataset to illustrate these effects (see *Video 1* for raw video). *Figure 1A* shows an example frame of the selected data, which contains large signals additional to the neurons visible in the focal plane. These extra fluorescence signals contribute as background that contaminates the single-neuronal signals of interest. In turn, standard methods based on local correlations for visualizing cell outlines (*Smith and Häusser, 2010*) are not effective here, because the correlations in the fluorescence of nearby pixels are dominated by background signals (*Figure 1B*). For some neurons with strong visible signals, we can manually draw regions-of-interest (ROI) (*Figure 1C*). Following (*Barbera et al., 2016*; *Pinto and Dan, 2015*), we used the mean fluorescence trace of the surrounding pixels (blue, *Figure 1D*) to roughly estimate this background fluctuation; subtracting it from the raw trace in the neuron ROI yields a relatively good estimation of neuron signal (red, *Figure 1D*). *Figure 1D* shows that the background (blue) has much larger variance than the relatively sparse neural signal (red); moreover, the background signal fluctuates on similar timescales as the single-neuronal signal, so we can not simply temporally filter the background away after extraction of the mean signal within the ROI. This large background signal is likely due to a combination of local fluctuations resulting from out-of-focus fluorescence or neuropil activity, hemodynamics of blood vessels, and global fluctuations shared more broadly across the field of view (photo-bleaching effects, drifts in $z$ of the focal plane, etc.), as illustrated schematically in *Figure 1E*.

The existing methods for extracting individual neural activity from microendoscopic data can be divided into two classes: semi-manual ROI analysis (*Barbera et al., 2016*; *Klaus et al., 2017*; *Pinto and Dan, 2015*) and PCA/ICA analysis (*Mukamel et al., 2009*). Unfortunately, both approaches have well-known flaws (*Resendez et al., 2016*). For example, ROI analysis does not effectively demix signals of spatially overlapping neurons, and drawing ROIs is laborious for large population recordings. More importantly, in many cases, the background contaminations are not adequately corrected, and thus the extracted signals are not sufficiently clean enough for downstream analyses. As for PCA/ICA analysis, it is a linear demixing method and therefore typically fails when the neural components exhibit strong spatial overlaps (*Pnevmatikakis et al., 2016*), as is the case in the microendoscopic setting.

Recently, constrained nonnegative matrix factorization (CNMF) approaches were proposed to simultaneously denoise, deconvolve, and demix calcium imaging data (*Pnevmatikakis et al., 2016*). However, current implementations of the CNMF approach were optimized for 2-photon and light-

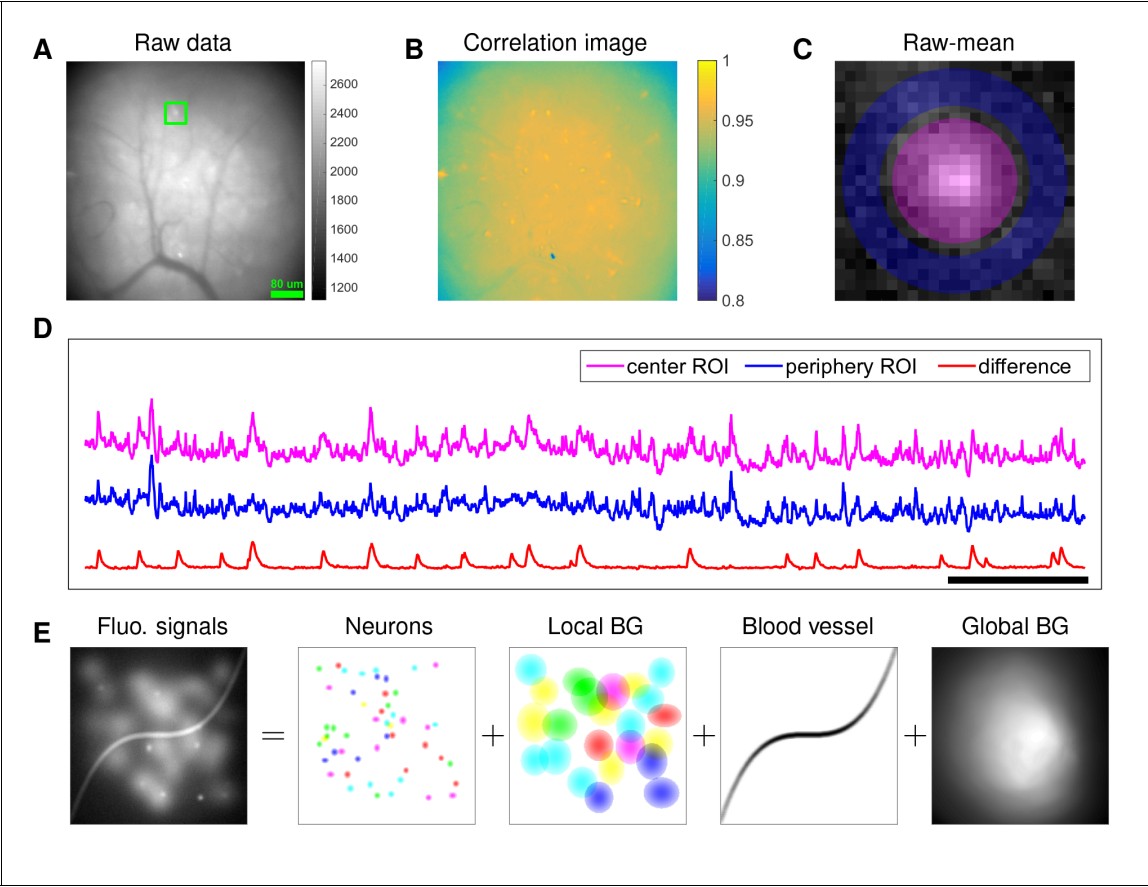

**Figure 1.** Microendoscopic data contain large background signals with rapid fluctuations due to multiple sources. (**A**) An example frame of microendoscopic data recorded in dorsal striatum (see Materials and methods section for experimental details). (**B**) The local 'correlation image' (*Smith and Häusser, 2010*) computed from the raw video data. Note that it is difficult to discern neuronal shapes in this image due to the high background spatial correlation level. (**C**) The mean-subtracted data within the cropped area (green) in (**A**). Two ROIs were selected and coded with different colors. (**D**) The mean fluorescence traces of pixels within the two selected ROIs (magenta and blue) shown in (**C**) and the difference between the two traces. (**E**) Cartoon illustration of various sources of fluorescence signals in microendoscopic data. 'BG' abbreviates 'background'.

DOI: https://doi.org/10.7554/eLife.28728.002

sheet microscopy, where the background has a simpler spatiotemporal structure. When applied to microendoscopic data, CNMF often has poor performance because the background is not modeled sufficiently accurately (*Barbera et al., 2016*).

In this paper, we significantly extend the CNMF framework to obtain a robust approach for extracting single-neuronal signals from microendoscopic data. Specifically, our extended CNMF for microendoscopic data (CNMF-E) approach utilizes a more accurate and flexible spatiotemporal background model that is able to handle the properties of the strong background signal illustrated in *Figure 1*, along with new specialized algorithms to initialize and fit the model components. After a brief description of the model and algorithms, we first use simulated data to illustrate the power of the new approach. Next, we compare CNMF-E with PCA/ICA analysis comprehensively on both simulated data and four experimental datasets recorded in different brain areas. The results show that CNMF-E outperforms PCA/ICA in terms of detecting more well-isolated neural signals, extracting higher signal-to-noise ratio (SNR) cellular signals, and obtaining more robust results in low SNR regimes. Finally, we show that downstream analyses of calcium imaging data can substantially benefit from these improvements.

## Model and model fitting

### CNMF for microendoscope data (CNMF-E)

The recorded video data can be represented by a matrix $Y \in \mathbb{R}_+^{d \times T}$, where $d$ is the number of pixels in the field of view and $T$ is the number of frames observed. In our model, each neuron $i$ is characterized by its spatial 'footprint' vector $a_i \in \mathbb{R}_+^d$ characterizing the cell's shape and location, and 'calcium activity' timeseries $c_i \in \mathbb{R}_+^T$, modeling (up to a multiplicative and additive constant) cell $i$'s mean fluorescence signal at each frame. Here, both $a_i$ and $c_i$ are constrained to be nonnegative because of their physical interpretations. The background fluctuation is represented by a matrix $B \in \mathbb{R}_+^{d \times T}$. If the field of view contains a total number of $K$ neurons, then the observed movie data is modeled as a superposition of all neurons' spatiotemporal activity, plus time-varying background and additive noise:

$$Y = \sum_{i=1}^K a_i \cdot c_i^T + B + E = AC + B + E, \tag{1}$$

where $A = [a_1, \ldots, a_K]$ and $C = [c_1, \ldots, c_K]^T$. The noise term $E \in \mathbb{R}^{d \times T}$ is modeled as Gaussian, $E(t) \sim \mathcal{N}(\boldsymbol{0}, \Sigma)$ is a diagonal matrix, indicating that the noise is spatially and temporally uncorrelated.

Estimating the model parameters $A, C$ in model (1) gives us all neurons' spatial footprints and their denoised temporal activity. This can be achieved by minimizing the residual sum of squares (RSS), aka the Frobenius norm of the matrix $Y - (AC + B)$,

$$\|Y - (AC + B)\|_F^2, \tag{2}$$

while requiring the model variables $A, C$ and $B$ to follow the desired constraints, discussed below.

## Constraints on neuronal spatial footprints $A$ and neural temporal traces $C$

Each spatial footprint $a_i$ should be spatially localized and sparse, since a given neuron will cover only a small fraction of the field of view, and therefore most elements of $a_i$ will be zero. Thus, we need to incorporate spatial locality and sparsity constraints on $A$ (*Pnevmatikakis et al., 2016*). We discuss details further below.

Similarly, the temporal components $c_i$ are highly structured, as they represent the cells' fluorescence responses to sparse, nonnegative trains of action potentials. Following (*Vogelstein et al., 2010*; *Pnevmatikakis et al., 2016*), we model the calcium dynamics of each neuron $c_i$ with a stable autoregressive (AR) process of order $p$,

$$c_i(t) = \sum_{j=1}^p \gamma_j^{(i)} c_i(t-j) + s_i(t), \tag{3}$$

where $s_i(t) \geq 0$ is the number of spikes that neuron fired at the $t$-th frame. (Note that there is no further noise input into $c_i(t)$ beyond the spike signal $s_i(t)$.) The AR coefficients $\{\gamma_j^{(i)}\}$ are different for each neuron and they are estimated from the data. In practice, we usually pick $p = 2$, thus incorporating both a nonzero rise and decay time of calcium transients in response to a spike; then *Equation (3)* can be expressed in matrix form as

$$G_i \cdot c_i = s_i, \text{ with } G_i = \begin{bmatrix} 1 & 0 & 0 & \cdots & 0 \\ -\gamma_1^{(i)} & 1 & 0 & \cdots & 0 \\ -\gamma_2^{(i)} & -\gamma_1^{(i)} & 1 & \cdots & 0 \\ \vdots & \ddots & \ddots & \ddots & \vdots \\ 0 & \cdots & -\gamma_2^{(i)} & -\gamma_1^{(i)} & 1 \end{bmatrix}. \tag{4}$$

The neural activity $s_i$ is nonnegative and typically sparse; to enforce sparsity, we can penalize the $\ell_0$ (*Jewell and Witten, 2017*) or $\ell_1$ (*Pnevmatikakis et al., 2016*; *Vogelstein et al., 2010*) norm of $s_i$, or limit the minimum size of nonzero spike counts (*Friedrich et al., 2017b*). When the rise time

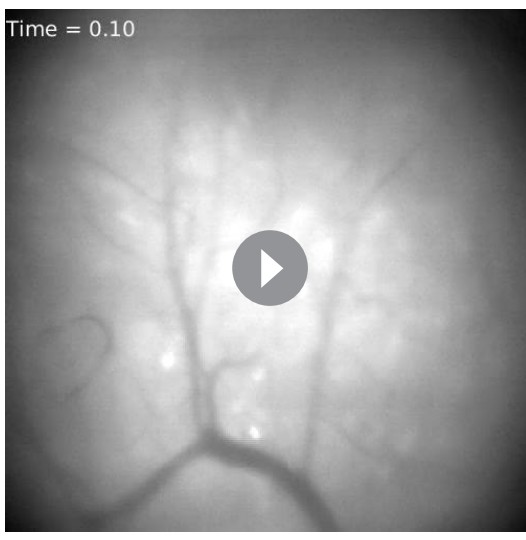

**Video 1.** An example of typical microendoscopic data. The video was recorded in dorsal striatum; experimental details can be found above. MP4
DOI: https://doi.org/10.7554/eLife.28728.003

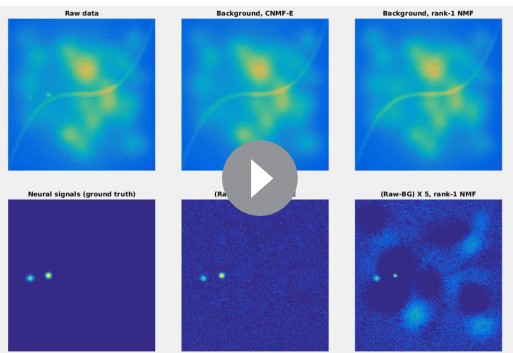

**Video 2.** Comparison of CNMF-E with rank-1 NMF in estimating background fluctuation in simulated data. Top left: the simulated fluorescence data in *Figure 2*. Bottom left: the ground truth of neuron signals in the simulation. Top middle: the estimated background from the raw video data (top left) using CNMF-E. Bottom middle: the residual of the raw video after subtracting the background estimated with CNMF-E. Top right and top bottom: same as top middle and bottom middle, but the background is estimated with rank-1 NMF. MP4
DOI: https://doi.org/10.7554/eLife.28728.005

constant is small compared to the timebin width (low imaging frame rate), we typically use a simpler AR(1) model (with an instantaneous rise following a spike) (*Pnevmatikakis et al., 2016*).

## Constraints on background activity $B$

In the above we have largely followed previously described CNMF approaches (*Pnevmatikakis et al., 2016*) for modeling calcium imaging signals. However, to accurately model the background effects in microendoscopic data, we need to depart significantly from these previous approaches. Constraints on the background term $B$ in *Equation (1)* are essential to the success of CNMF-E, since clearly, if $B$ is completely unconstrained we could just absorb the observed data $Y$ entirely into $B$, which would lead to recovery of no neural activity. At the same time, we need to prevent the residual of the background term (i.e. $B - \hat{B}$, where $\hat{B}$ denotes the estimated spatiotemporal background) from corrupting the estimated neural signals $AC$ in model (1), since subsequently, the extracted neuronal activity would be mixed with background fluctuations, leading to artificially high correlations between nearby cells. This problem is even worse in the microendoscopic context because the background fluctuation usually has significantly larger variance than the isolated cellular signals of interest (*Figure 1D*), and therefore any small errors in the estimation of $B$ can severely corrupt the estimated neural signal $AC$.

In (*Pnevmatikakis et al., 2016*), $B$ is modeled as a rank-1 nonnegative matrix $B = b \cdot f^T$, where $b \in \mathbb{R}_+^d$ and $f \in \mathbb{R}_+^T$. This model mainly captures the global fluctuations within the field of view (FOV). In applications to two-photon or light-sheet data, this rank-1 model has been shown to be sufficient for relatively small spatial regions; the simple low-rank model does not hold for larger fields of view, and so we can simply divide large FOVs into smaller patches for largely parallel processing (*Pnevmatikakis et al., 2016*; *Giovannucci et al., 2017b*). (See [*Pachitariu et al., 2016*] for an alternative approach.) However, as we will see below, the local rank-1 model fails in many microendoscopic datasets, where multiple large overlapping background sources exist even within modestly sized FOVs.

Thus, we propose a new model to constrain the background term $B$. We first decompose the background into two terms:

$$B = B^f + B^c, \tag{5}$$

where $B^f$ represents fluctuating activity and $B^c = b_0 \cdot 1^T$ models constant baselines ($1 \in \mathbb{R}^T$ denotes a vector of $T$ ones). To model $B^f$, we exploit the fact that background sources (largely due to blurred out-of-focus fluorescence) are empirically much coarser spatially than the average neuron soma size $l$. Thus, we model $B^f$ at one pixel as a linear combination of the background fluorescence in pixels which are chosen to be nearby but not nearest neighbors:

$$B^f_{it} = \sum_{j \in \Omega_i} w_{ij} \cdot B^f_{jt}, \ \forall t = 1 \ldots T, \tag{6}$$

where $\Omega_i = \{j \mid \mathrm{dist}(\boldsymbol{x}_i, \boldsymbol{x}_j) \in [l_n, l_n + 1)\}$, with $\mathrm{dist}(x_i, x_j)$ the Euclidean distance between pixel $i$ and $j$. Thus, $\Omega_i$ only selects the neighboring pixels with a distance of $l_n$ from the $i$-th pixel (the green dot and black pixels in *Figure 2B* illustrate $i$ and $\Omega_i$, respectively); here $l_n$ is a parameter that we choose to be greater than $l$ (the size of the typical soma in the FOV), e.g., $l_n = 2l$. This choice of $l_n$ ensures that pixels $i$ and $j$ in *Equation (6)* share similar background fluctuations, but do not belong to the same soma.

We can rewrite *Equation (6)* in matrix form:

$$B^f = WB^f, \tag{7}$$

where $W_{ij} = 0$ if $\mathrm{dist}(x_i, x_j) \notin [l_n, l_n + 1)$. In practice, this hard constraint is difficult to enforce computationally and is overly stringent given the noisy observed data. We relax the model by replacing the right-hand side $B^f$ with the more convenient closed-form expression

$$B^f = W \cdot (Y - AC - b_0 \cdot 1^T). \tag{8}$$

According to *Equations (1) and (5)*, this change ignores the noise term $E$; since elements in $E$ are spatially uncorrelated, $W \cdot E$ contributes as a very small disturbance to $\hat{B}^f$ in the left-hand side. We found this substitution for $\hat{B}^f$ led to significantly faster and more robust model fitting.

## Fitting the CNMF-E model

*Table 1* lists the variables in the proposed CNMF-E model. Now we can formulate the estimation of all model variables as a single optimization meta-problem:

$$\begin{aligned}
\underset{A,C,S,B^f,W,\boldsymbol{b}_0}{\text{minimize}} \quad & \|Y - AC - \boldsymbol{b}_0 \cdot \boldsymbol{1}^T - B^f\|_F^2 \\
\text{subject to} \quad & A \geq 0, \ A \text{ is sparse and spatially localized} \\
& \boldsymbol{c}_i \geq 0, \ \boldsymbol{s}_i \geq 0, \ G^{(i)}\boldsymbol{c}_i = \boldsymbol{s}_i, \boldsymbol{s}_i \text{ is sparse } \forall i = 1 \ldots K \\
& B^f \cdot \boldsymbol{1} = \boldsymbol{0} \\
& B^f = W \cdot (Y - AC - \boldsymbol{b}_0 \cdot \boldsymbol{1}^T) \\
& W_{ij} = 0 \text{ if } \mathrm{dist}(\boldsymbol{x}_i, \boldsymbol{x}_j) \notin [l_n, l_n + 1).
\end{aligned} \tag{P-All}$$

We call this a 'meta-problem' because we have not yet explicitly defined the sparsity and spatial locality constraints on $A$ and $S = [s_1, \ldots, s_K]^T$; these can be customized by users under different assumptions (see details in Materials and methods). Also note that $s_i$ is completely determined by $c_i$ and $G^{(i)}$, and $B^f$ is not optimized explicitly but (as discussed above) can be estimated as $W \cdot (Y - AC - b_0 \cdot 1^T)$, so we optimize with respect to $W$ instead.

The problem (P-All) optimizes all variables together and is non-convex but can be divided into three simpler subproblems that we solve iteratively:

Estimating $A, b_0$ given $\hat{C}, \hat{B}^f$

$$\begin{aligned}
\underset{A,\boldsymbol{b}_0}{\text{minimize}} \quad & \|Y - A \cdot \hat{C} - \boldsymbol{b}_0 \cdot \boldsymbol{1}^T - \hat{B}^f\|_F^2 \\
\text{subject to} \quad & A \geq 0, A \text{ is sparse and spatially localized}
\end{aligned} \tag{P-S}$$

Estimating $C, b_0$ given $\hat{A}, \hat{B}^f$

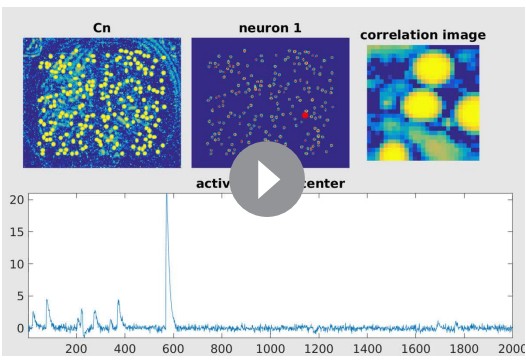

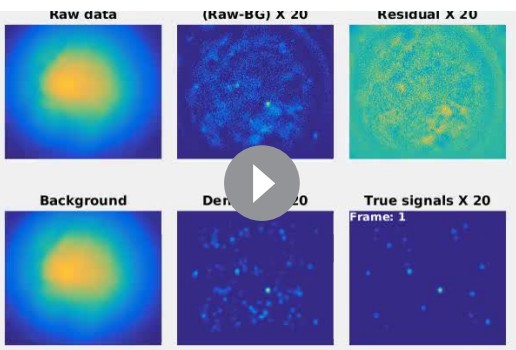

**Video 3.** Initialization procedure for the simulated data in *Figure 3*. Top left: correlation image of the filtered data. Red dots are centers of initialized neurons. Top middle: candidate seed pixels (small red dots) for initializing neurons on top of PNR image. The large red dot indicates the current seed pixel. Top right: the correlation image surrounding the selected seed pixel or the spatial footprint of the initialized neuron. Bottom: the filtered fluorescence trace at the seed pixel or the initialized temporal activity (both raw and denoised). MP4
DOI: https://doi.org/10.7554/eLife.28728.008

**Video 4.** The results of CNMF-E in demixing simulated data in *Figure 4* (SNR reduction factor = 1). Top left: the simulated fluorescence data. Bottom left: the estimated background. Top middle: the residual of the raw video (top left) after subtracting the estimated background (bottom left). Bottom middle: the denoised neural signals. Top right: the residual of the raw video data (top right) after subtracting the estimated background (bottom left) and denoised neural signal (bottom middle). Bottom right: the ground truth of neural signals in simulation. MP4
DOI: https://doi.org/10.7554/eLife.28728.010

$$\underset{C,S,b_0}{\text{minimize}} \quad \|Y - \hat{A} \cdot C - \boldsymbol{b}_0 \cdot \boldsymbol{1}^T - \hat{B}^f\|_F^2 \qquad \text{(P-T)}$$
$$\text{subject to} \quad \boldsymbol{c}_i \geq 0, \boldsymbol{s}_i \geq 0$$
$$G^{(i)}\boldsymbol{c}_i = \boldsymbol{s}_i, \boldsymbol{s}_i \text{ is sparse } \forall i = 1 \ldots K$$

Estimating $W, b_0$ given $\hat{A}, \hat{C}$

$$\underset{W,B^f,\boldsymbol{b}_0}{\text{minimize}} \quad \|Y - \hat{A} \cdot \hat{C} - \boldsymbol{b}_0 \cdot \boldsymbol{1}^T - B^f\|_F^2 \qquad \text{(P-B)}$$
$$\text{subject to} \quad B^f \cdot \boldsymbol{1} = \boldsymbol{0}$$
$$B^f = W \cdot (Y - \hat{A} \cdot \hat{C} - \boldsymbol{b}_0 \cdot \boldsymbol{1}^T).$$
$$W_{ij} = 0 \text{ if } \text{dist}(\boldsymbol{x}_i, \boldsymbol{x}_j) \notin [l_n, l_n + 1)$$

For each of these subproblems, we are able to use well-established algorithms (e.g. solutions for (P-S) and (P-T) are discussed in *Friedrich et al., 2017a*; *Pnevmatikakis et al., 2016*) or slight modifications thereof. By iteratively solving these three subproblems, we obtain tractable updates for all model variables in problem (P-All). Furthermore, this strategy gives us the flexibility of further potential interventions (either automatic or semi-manual) in the optimization procedure, for example, incorporating further prior information on neurons' morphology, or merging/splitting/deleting spatial components and detecting missed neurons from the residuals. These steps can significantly improve the quality of the model fitting; this is an advantage compared with PCA/ICA, which offers no easy option for incorporation of stronger prior information or manually guided improvements on the estimates.

Full details on the algorithms for initializing and then solving these three subproblems are provided in the Materials and methods section.

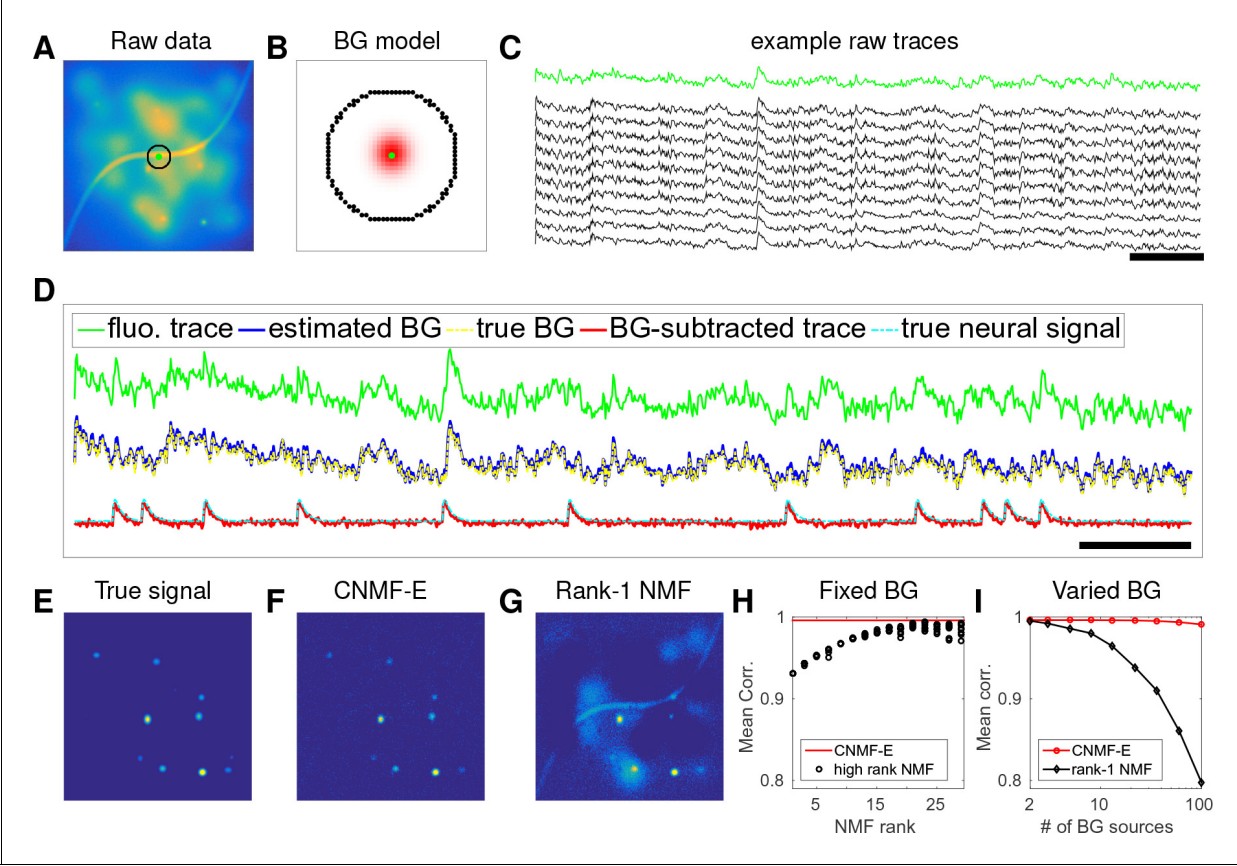

**Figure 2.** CNMF-E can accurately separate and recover the background fluctuations in simulated data. (**A**) An example frame of simulated microendoscopic data formed by summing up the fluorescent signals from the multiple sources illustrated in *Figure 1E*. (**B**) A zoomed-in version of the circle in (**A**). The green dot indicates the pixel of interest. The surrounding black pixels are its neighbors with a distance of 15 pixels. The red area approximates the size of a typical neuron in the simulation. (**C**) Raw fluorescence traces of the selected pixel and some of its neighbors on the black ring. Note the high correlation. (**D**) Fluorescence traces (raw data; true and estimated background; true and initial estimate of neural signal) from the center pixel as selected in (**B**). Note that the background dominates the raw data in this pixel, but nonetheless we can accurately estimate the background and subtract it away here. Scalebars: 10 seconds. Panels (**E–G**) show the cellular signals in the same frame as (**A**). (**E**) Ground truth neural activity. (**F**) The residual of the raw frame after subtracting the background estimated with CNMF-E; note the close correspondence with E. (**G**) Same as (**F**), but the background is estimated with rank-1 NMF. A video showing (**E–G**) for all frames can be found at *Video 2*. (**H**) The mean correlation coefficient (over all pixels) between the true background fluctuations and the estimated background fluctuations. The rank of NMF varies and we run randomly-initialized NMF for 10 times for each rank. The red line is the performance of CNMF-E, which requires no selection of the NMF rank. (**I**) The performance of CNMF-E and rank-1 NMF in recovering the background fluctuations from the data superimposed with an increasing number of background sources.
DOI: https://doi.org/10.7554/eLife.28728.004

## Results

### CNMF-E can reliably estimate large high-rank background fluctuations

We first use simulated data to illustrate the background model in CNMF-E and compare its performance against the low-rank NMF model used in the basic CNMF approach (*Pnevmatikakis et al., 2016*). We generated the observed fluorescence $Y$ by summing up simulated fluorescent signals of multiple sources as shown in *Figure 1E* plus additive Gaussian white noise (*Figure 2A*).

An example pixel (green dot, *Figure 2A,B*) was selected to illustrate the background model in CNMF-E (*Equation (6)*), which assumes that each pixel's background activity can be reconstructed using its neighboring pixels' activities. The selected neighbors form a ring and their distances to the center pixel are larger than a typical neuron size (*Figure 2B*). *Figure 2C* shows that the fluorescence traces of the center pixel and its neighbors are highly correlated due to the shared large background fluctuations. Here, for illustrative purposes, we fit the background by solving problem (P-B) directly

while assuming $\hat{A}\hat{C} = 0$. This mistaken assumption should make the background estimation more challenging (due to true neural components getting absorbed into the background), but nonetheless in *Figure 2* we see that the background fluctuation was well recovered (*Figure 2D*). Subtracting this estimated background from the observed fluorescence in the center yields a good visualization of the cellular signal (*Figure 2D*). Thus, this example shows that we can reconstruct a complicated background trace while leaving the neural signal uncontaminated.

For the example frame in *Figure 2A*, the true cellular signals are sparse and weak (*Figure 2E*). When we subtract the estimated background using CNMF-E from the raw data, we obtain a good recovery of the true signal (*Figure 2D,F*). For comparison, we also estimate the background activity by applying a rank-1 NMF model as used in basic CNMF; the resulting background-subtracted image is still severely contaminated by the background (*Figure 2G*). This is easy to understand: the spatio-temporal background signal in microendoscopic data typically has a rank higher than one, due to the various signal sources indicated in *Figure 1E*), and therefore a rank-1 NMF background model is insufficient.

A naive approach would be to simply increase the rank of the NMF background model. *Figure 2H* demonstrates that this approach is ineffective: higher rank NMF does yield generally better reconstruction performance, but with high variability and low reliability (due to randomness in the initial conditions of NMF). Eventually as the NMF rank increases many single-neuronal signals of interest are swallowed up in the estimated background signal (data not shown). In contrast, CNMF-E recovers the background signal more accurately than any of the high-rank NMF models.

In real data analysis settings, the rank of NMF is an unknown and the selection of its value is a nontrivial problem. We simulated data sets with different numbers of local background sources and use a single parameter setting to run CNMF-E for reconstructing the background over multiple such simulations. *Figure 2I* shows that the performance of CNMF-E does not degrade quickly as we have more background sources, in contrast to rank-1 NMF. Therefore, CNMF-E can recover the background accurately across a diverse range of background sources, as desired.

## CNMF-E accurately initializes single-neuronal spatial and temporal components

Next, we used simulated data to validate our proposed initialization procedure (*Figure 3A*). In this example, we simulated 200 neurons with strong spatial overlaps (*Figure 3B*). One of the first steps in our initialization procedure is to apply a Gaussian spatial filter to the images to reduce the (spatially coarser) background and boost the power of neuron-sized objects in the images. In *Figure 3C*, we see that the local correlation image (*Smith and Häusser, 2010*) computed on the spatially filtered data provides a good initial visualization of neuron locations; compare to *Figure 1B*, where the correlation image computed on the raw data was highly corrupted by background signals.

We choose two example ROIs to illustrate how CNMF-E removes the background contamination and demixes nearby neural signals for accurate initialization of neurons' shapes and activity. In the first example, we choose a well-isolated neuron (green box, *Figure 3A+B*). We select three pixels located in the center, the periphery, and the outside of the neuron and show the corresponding fluorescence traces in both the raw data and the spatially filtered data (*Figure 3D*). The raw traces are noisy and highly correlated, but the filtered traces show relatively clean neural signals. This is because spatial filtering reduces the shared background activity and the remaining neural signals dominate the filtered data. Similarly, *Figure 3E* is an example showing how CNMF-E demixes two overlapping neurons. The filtered traces in the centers of the two neurons still preserve their own temporal activity.

After initializing the neurons' traces using the spatially filtered data, we initialize our estimate of

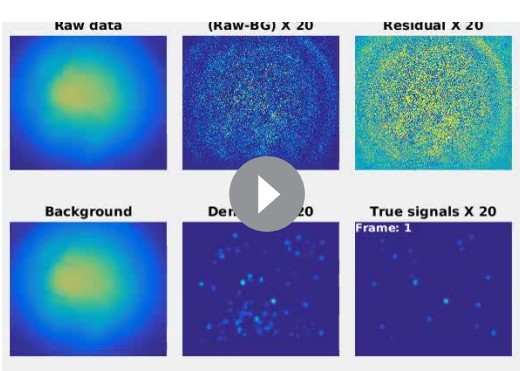

**Video 5.** The results of CNMF-E in demixing the simulated data in *Figure 4* (SNR reduction factor = 6). Conventions as in previous video. MP4
DOI: https://doi.org/10.7554/eLife.28728.011

their spatial footprints. Note that simply initializing these spatial footprints with the spatially filtered data does not work well (data not shown), since the resulting shapes are distorted by the spatial filtering process. We found that it was more effective to initialize each spatial footprint by regressing the initial neuron traces onto the raw movie data (see Materials and methods for details). The initial values already match the simulated ground truth with fairly high fidelity (*Figure 3D+E*). In this simulated data, CNMF-E successfully identified all 200 neurons and initialized their spatial and temporal components (*Figure 3F*). We then evaluate the quality of initialization using all neurons' spatial and temporal similarities with their counterparts in the ground truth data. *Figure 3G* shows that all initialized neurons have high similarities with the truth, indicating a good recovery and demixing of all neuron sources.

Thresholds on the minimum local correlation and the minimum peak-to-noise ratio (PNR) for detecting seed pixels are necessary for defining the initial spatial components. To quantify the sensitivity of choosing these two thresholds, we plot the local correlations and the PNRs of all pixels chosen as the local maxima within an area of $\frac{l}{4} \times \frac{l}{4}$, where $l$ is the diameter of a typical neuron, in the correlation image or the PNR image (*Figure 3H*). Pixels are classified into two classes according to their locations relative to the closest neurons: neurons' central areas and outside areas (see Materials and methods for full details). It is clear that the two classes are linearly well separated and the thresholds can be chosen within a broad range of values (*Figure 3H*), indicating that the algorithm is robust with respect to these threshold parameters here. In lower SNR settings, these boundaries may be less clear, and an incremental approach (in which we choose the highest-SNR neurons first, then estimate the background and examine the residual to select the lowest-SNR cells) may be preferred; this incremental approach is discussed in more depth in the Materials and methods section.

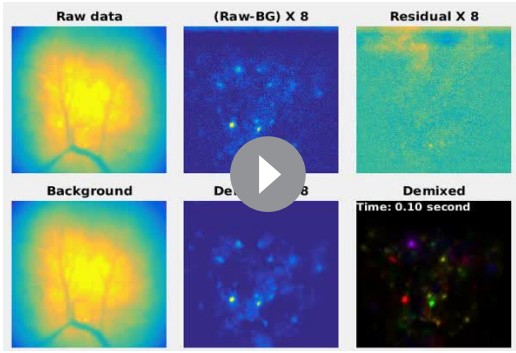

**Video 6.** The results of CNMF-E in demixing dorsal striatum data. Top left: the recorded fluorescence data. Bottom left: the estimated background. Top middle: the residual of the raw video (top left) after subtracting the estimated background (bottom left). Bottom middle: the denoised neural signals. Top right: the residual of the raw video data (top right) after subtracting the estimated background (bottom left) and denoised neural signal (bottom middle). Bottom right: the denoised neural signals while all neurons' activity are coded with pseudocolors. MP4
DOI: https://doi.org/10.7554/eLife.28728.014

## CNMF-E recovers the true neural activity and is robust to noise contamination and neuronal correlations in simulated data

Using the same simulated dataset as in the previous section, we further refine the neuron shapes ($A$) and the temporal traces ($C$) by iteratively fitting the CNMF-E model. We compare the final results with PCA/ICA analysis (*Mukamel et al., 2009*) and the original CNMF method (*Pnevmatikakis et al., 2016*).

After choosing the thresholds for seed pixels (*Figure 3H*), we run CNMF-E in full automatic mode, without any manual interventions. Two open-source MATLAB packages, CellSort (https://github.com/mukamel-lab/CellSort; *Mukamel, 2016*) and ca_source_extraction (https://github.com/epnev/ca_source_extraction; *Pnevmatikakis, 2016*), were used to perform PCA/ICA (*Mukamel et al., 2009*) and basic CNMF (*Pnevmatikakis et al., 2016*), respectively. Since the initialization algorithm in CNMF fails due to the large contaminations from the background fluctuations in this setting (recall *Figure 2*), we use the ground truth as its initialization. As for the rank of the background model in CNMF, we tried all integer values between 1 and 16 and set it as 7 because it has the best performance in matching the ground truth. We emphasize that including the CNMF approach in this comparison is not fair for the other two approaches, because it uses the ground truth heavily, while PCA/ICA and CNMF-E are blind to the ground truth. The purpose here is to show the limitations of basic CNMF in modeling the background activity in microendoscopic data.

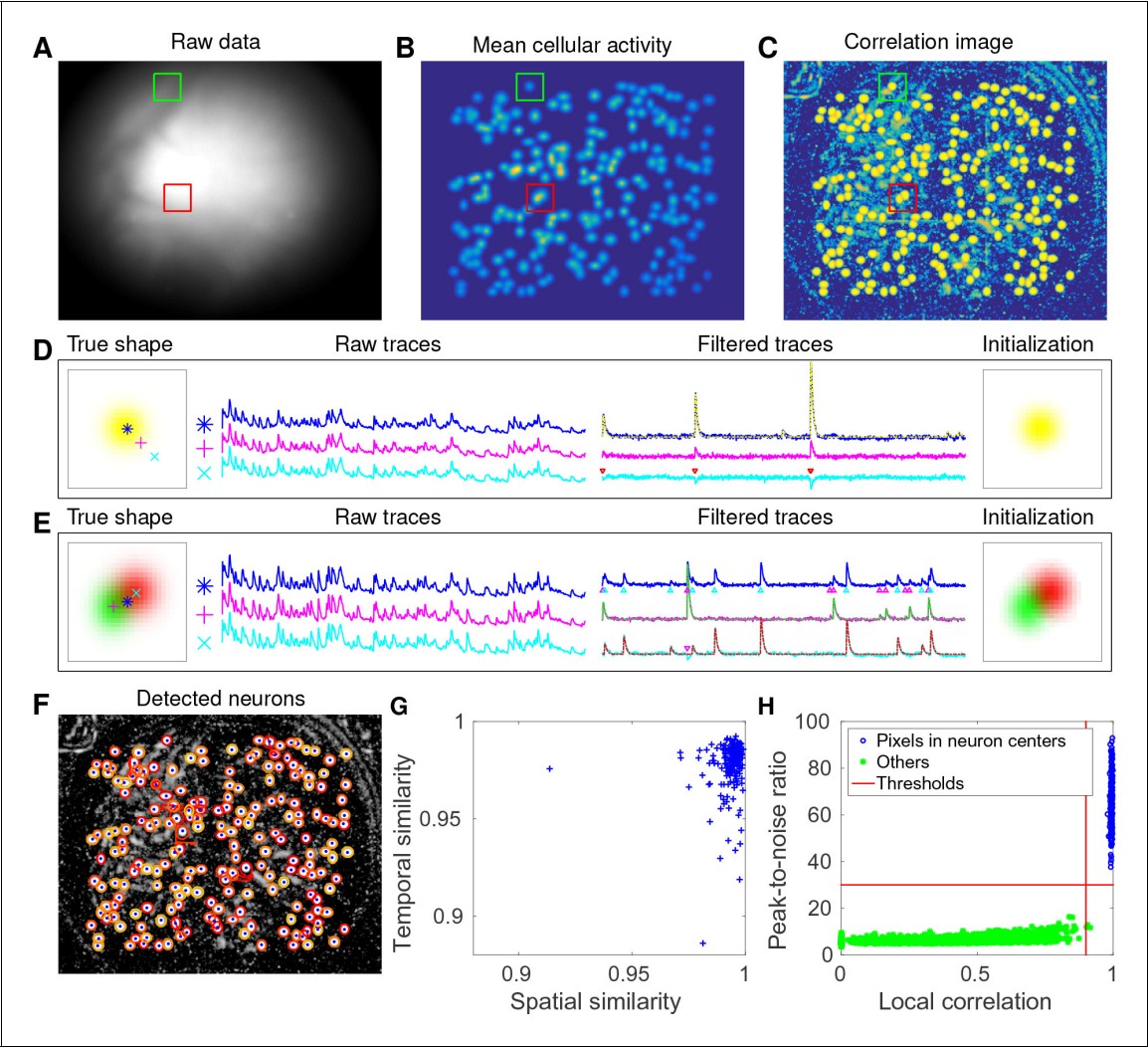

**Figure 3.** CNMF-E accurately initializes individual neurons' spatial and temporal components in simulated data. (A) An example frame of the simulated data. Green and red squares will correspond to panels (D) and (E) below, respectively. (B) The temporal mean of the cellular activity in the simulation. (C) The correlation image computed using the spatially filtered data. (D) An example of initializing an isolated neuron. Three selected pixels correspond to the center, the periphery, and the outside of a neuron. The raw traces and the filtered traces are shown as well. The yellow dashed line is the true neural signal of the selected neuron. Triangle markers highlight the spike times from the neuron. (E) Same as (D), but two neurons are spatially overlapping in this example. Note that in both cases neural activity is clearly visible in the filtered traces, and the initial estimates of the spatial footprints are already quite accurate (dashed lines are ground truth). (F) The contours of all initialized neurons on top of the correlation image as shown in (D). Contour colors represent the rank of neurons' SNR (SNR decreases from red to yellow). The blue dots are centers of the true neurons. (G) The spatial and the temporal cosine similarities between each simulated neuron and its counterpart in the initialized neurons. (H) The local correlation and the peak-to-noise ratio for pixels located in the central area of each neuron (blue) and other areas (green). The red lines are the thresholding boundaries for screening seed pixels in our initialization step. A video showing the whole initialization step can be found at *Video 3*.
DOI: https://doi.org/10.7554/eLife.28728.007

We first pick three closeby neurons from the ground truth (*Figure 4A*, top) and see how well these neurons' activities are recovered. PCA/ICA fails to identify one neuron, and for the other two identified neurons, it recovers temporal traces that are sufficiently noisy that small calcium transients are submerged in the noise. As for CNMF, the neuron shapes remain more or less at the initial condition (i.e. the ground truth spatial footprints), but clear contaminations in the temporal traces are visible. This is because the pure NMF model in CNMF does not model the true background well and the residuals in the background are mistakenly captured by neural components. In contrast, on this example, CNMF-E recovers the true neural shapes and neural activity with high accuracy.

We also compare the number of detected neurons: PCA/ICA detected 195 out of 200 neurons, while CNMF-E detected all 200 neurons. We also quantitatively evaluated the performance of source extraction by showing the spatial and temporal cosine similarities between detected neurons and ground truth (*Figure 4C*); we find that the neurons detected using PCA/ICA have much lower similarities with the ground truth (*Figure 4C*). We also note that the CNMF results are much worse than those of CNMF-E here, despite the fact that CNMF is initialized at the ground truth parameter values. This result clarifies an important point: the improvements from CNMF-E are not simply due to improvements in the initialization step. Furthermore, running the full iterative pipeline of CNMF-E leads to improvements in both spatial and temporal similarities, compared with the results in the initialization step.

In many downstream analyses of calcium imaging data, pairwise correlations provide an important metric to study coordinated network activity (*Warp et al., 2012*; *Barbera et al., 2016*; *Dombeck et al., 2009*; *Klaus et al., 2017*). Since PCA/ICA seeks statistically independent components, which forces the temporal traces to have near-zero correlation, the correlation structure is badly corrupted in the raw PCA/ICA outputs (*Figure 4D*). We observed that a large proportion of the independence comes from the noisy baselines in the extracted traces (data not shown), so we postprocessed the PCA/ICA output by thresholding at the 3 standard deviation level. This recovers some nonzero correlations, but the true correlation structure is not recovered accurately (*Figure 4D*). By contrast, the CNMF-E results matched the ground truth very well due to accurate extraction of individual neurons' temporal activity (*Figure 4D*). As for CNMF, the estimated correlations are slightly elevated relative to the true correlations. This is due to the shared (highly correlated) background fluctuations that corrupt the recovered activity of nearby neurons.

Next, we compared the performance of the different methods under different SNR regimes. Because of the above inferior results we skip comparisons to the basic CNMF here. Based on the same simulation parameters as above, we vary the noise level $\Sigma$ by multiplying it with a SNR reduction factor. *Figure 4E* shows that CNMF-E detects all neurons over a wide SNR range, while PCA/ICA fails to detect the majority of neurons when the SNR drops to sufficiently low levels. Moreover, the detected neurons in CNMF-E preserve high spatial and temporal similarities with the ground truth (*Figure 4F–G*). This high accuracy of extracting neurons' temporal activity benefits from the modeling of the calcium dynamics, which leads to significantly denoised neural activity. If we skip the temporal denoising step in the algorithm, CNMF-E is less robust to noise, but still outperforms PCA/ICA significantly (*Figure 4G*). When SNR is low, the improvements yielded by CNMF-E can be crucial for detecting weak neuron events, as shown in *Figure 4H*.

Finally, we examine the ability of CNMF-E to demix correlated and overlapping neurons. Using the two example neurons in *Figure 3E*, we ran multiple simulations at varying correlation levels and extracted neural components using the CNMF-E pipeline and PCA/ICA analysis. The spatial footprints in these simulations were fixed, but the temporal components were varied to have different correlation levels ($\gamma$) between calcium traces by tuning their shared component with the common background fluctuations. For high correlation levels ($\gamma > 0.7$), the initialization procedure tends to first initialize a component that explains the common activity between two neurons and then initialize another component to account for the residual of one neuron. After iteratively refining the model variables, CNMF-E successfully extracted the two neurons' spatiotemporal activity even at very high correlation levels ($\gamma = 0.95$; *Figure 5A,B*). PCA/ICA was also often able to separate two neurons for large correlation levels ($\gamma = 0.9$, *Figure 5B*), but the extracted traces have problematic negative spikes that serve to reduce their statistical dependences (*Figure 4A*).

## Application to dorsal striatum data

We now turn to the analysis of large-scale microendoscopic datasets recorded from freely behaving mice. We run both CNMF-E and PCA/ICA for all datasets and compare their performances in detail.

We begin by analyzing in vivo calcium imaging data of neurons expressing GCaMP6f in the mouse dorsal striatum. (Full experimental details and algorithm parameter settings for this and the following datasets appear in the Methods and Materials section.) CNMF-E extracted 692 putative neural components from this dataset; PCA/ICA extracted 547 components (starting from 700 initial components, and then automatically removing false positives using the same criterion as applied in CNMF-E). *Figure 6A* shows how CNMF-E decomposes an example frame into four components: the constant baselines that are invariant over time, the fluctuating background, the denoised neural

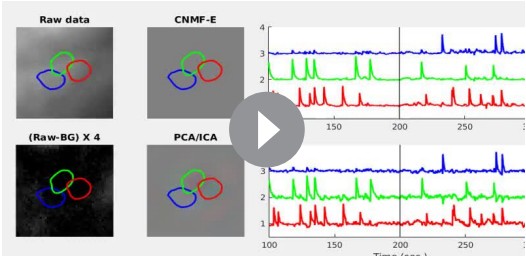

**Video 7.** The results of CNMF-E in demixing PFC data. Conventions as in previous video. MP4
DOI: https://doi.org/10.7554/eLife.28728.016

**Video 8.** Comparison of CNMF-E with PCA/ICA in demixing overlapped neurons in *Figure 7G*. Top left: the recorded fluorescence data. Bottom left: the residual of the raw video (top left) after subtracting the estimated background using CNMF-E. Top middle and top right: the spatiotemporal activity and temporal traces of three neurons extracted using CNMF-E. Bottom middle and bottom right: the spatiotemporal activity and temporal traces of three neurons extracted using PCA/ICA. MP4
DOI: https://doi.org/10.7554/eLife.28728.017

signals, and the residuals. We highlight an example neuron by drawing its ROI to demonstrate the power of CNMF-E in isolating fluorescence signals of neurons from the background fluctuations. For the selected neuron, we plot the mean fluorescence trace of the raw data and the estimated background (*Figure 6B*). These two traces are very similar, indicating that the background fluctuation dominates the raw data. By subtracting this estimated background component from the raw data, we acquire a clean trace that represents the neural signal.

To quantify the background effects further, we compute the contribution of each signal component in explaining the variance in the raw data. For each pixel, we compute the variance of the raw data first and then compute the variance of the background-subtracted data. Then the reduced variance is divided by the variance of the raw data, giving the proportion of variance explained by the background. *Figure 6C* (blue) shows the distribution of the background-explained variance over all pixels. The background accounts for around 90% of the variance on average. We further remove the denoised neural signals and compute the variance reduction; *Figure 6C* shows that neural signals account for less than 10% of the raw signal variance. This analysis is consistent with our observations that background dominates the fluorescence signal and extracting high-quality neural signals requires careful background signal removal.

The contours of the spatial footprints inferred by the two approaches (PCA/ICA and CNMF-E) are depicted in *Figure 6D*, superimposed on the correlation image of the filtered raw data. The indicated area was cropped from *Figure 6A* (left). In this case, most neurons inferred by PCA/ICA were inferred by CNMF-E as well, with the exception of a few components that seemed to be false positives (judging by their spatial shapes and temporal traces and visual inspection of the raw data movie; detailed data not shown). However, many realistic components were only detected by CNMF-E (shown as the green areas in *Figure 6D*). In these plots, we rank the inferred components according to their SNRs; the color indicates the relative rank (decaying from red to yellow). We see that the components missed by PCA/ICA have low SNRs (green shaded areas with yellow contours).

*Figure 6E* shows the spatial and temporal components of 14 example neurons detected only by CNMF-E. Here (and in the following figures), for illustrative purposes, we show the calcium traces before the temporal denoising step. For neurons that are inferred by both methods, CNMF-E shows significant improvements in the SNR of the extracted cellular signals (*Figure 6F*), even before the temporal denoising step is applied. In panel G we randomly select 10 examples and examine their spatial and temporal components. Compared with the CNMF-E results, PCA/ICA components have much smaller size, often with negative dips surrounding the neuron (remember that ICA avoids spatial overlaps in order to reduce nearby neurons' statistical dependences, leading to some loss of signal strength; see (*Pnevmatikakis et al., 2016*) for further discussion). The activity traces extracted by

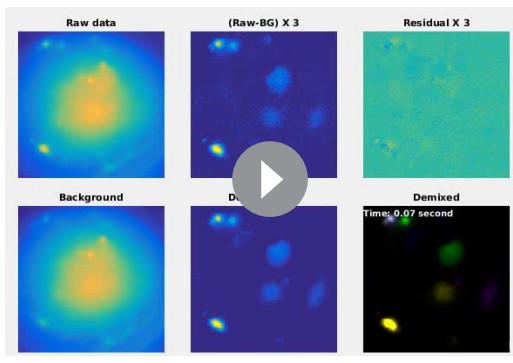

**Video 9.** The results of CNMF-E in demixing ventral hippocampus data. Conventions as in *Video 6*. MP4
DOI: https://doi.org/10.7554/eLife.28728.019

CNMF-E are visually cleaner than the PCA/ICA traces; this is important for reliable event detection, particularly in low SNR examples. See *Klaus et al., 2017*) for additional examples of CNMF-E applied to striatal data.

## Application to data in prefrontal cortex

We repeat a similar analysis on GCaMP6s data recorded from prefrontal cortex (PFC, *Figure 7*), to quantify the performance of the algorithm in a different brain area with a different calcium indicator. Again we find that CNMF-E successfully extracts neural signals from a strong fluctuating background (*Figure 7A*), which contributes a large proportion of the variance in the raw data (*Figure 7B*). Similarly as with the striatum data, PCA/ICA analysis missed many components that have very weak signals (33 missed components here). For the matched neurons, CNMF-E shows strong improvements in the SNRs of the extracted traces (*Figure 7D*). Consistent with our observation in striatum (*Figure 6G*), the spatial footprints of PCA/ICA components are shrunk to promote statistical independence between neurons, while the neurons inferred by CNMF-E have visually reasonable morphologies (*Figure 6E*). As for calcium traces with high SNRs (*Figure 7E*, cell 1-6), CNMF-E traces have smaller noise values, which is important for detecting small calcium transients (*Figure 7E*, cell 4). For traces with low SNRs (*Figure 7*, cell 7-10), it is challenging to detect any calcium events from the PCA/ICA traces due to the large noise variance; CNMF-E is able to visually recover many of these weaker signals. For those cells missed by PCA/ICA, their traces extracted by CNMF-E have reasonable morphologies and visible calcium events (*Figure 7F*).

The demixing performance of PCA/ICA analysis can be relatively weak because it is inherently a linear demixing method (*Pnevmatikakis et al., 2016*). Since CNMF-E uses a more suitable nonlinear matrix factorization method, it has a better capability of demixing spatially overlapping neurons. As an example, *Figure 7G* shows three closeby neurons identified by both CNMF-E and PCA/ICA analysis. PCA/ICA forces its obtained filters to be spatially separated to reduce their dependence (thus reducing the effective signal strength), while CNMF-E allows inferred spatial components to have large overlaps (*Figure 7G*, left), retaining the full signal power. In the traces extracted by PCA/ICA, the component labeled in green contains many negative 'spikes,' which are highly correlated with the spiking activity of the blue neuron (*Figure 7G*, yellow). In addition, the green PCA/ICA neuron has significant crosstalk with the red neuron due to the failure of signal demixing (*Figure 7G*, cyan); the CNMF-E traces shows no comparable negative 'spikes' or crosstalk. See also *Video 8* for further details.

## Application to ventral hippocampus neurons

In the previous two examples, we analyzed data with densely packed neurons, in which the neuron sizes are all similar. In the next example, we apply CNMF-E to a dataset with much sparser and more heterogeneous neural signals. The data used here were recorded from amygdala-projecting neurons expressing GCaMP6f in ventral hippocampus. In this dataset, some neurons that are slightly above or below the focal plane were visible with prominent signals, though their spatial shapes are larger than neurons in the focal plane.

This example is somewhat more challenging due to the large diversity of neuron sizes. It is possible to set multiple parameters to detect neurons of different sizes (or to e.g. differentially detect somas versus smaller segments of axons or dendrites passing through the focal plane), but for illustrative purposes here we use a single neural size parameter to initialize all of the components. This in turn splits some large neurons into multiple components. Following this crude initialization step, we updated the background component and then picked the missing neurons from the residual using a second greedy component initialization step. Next, we ran CNMF-E for three iterations of

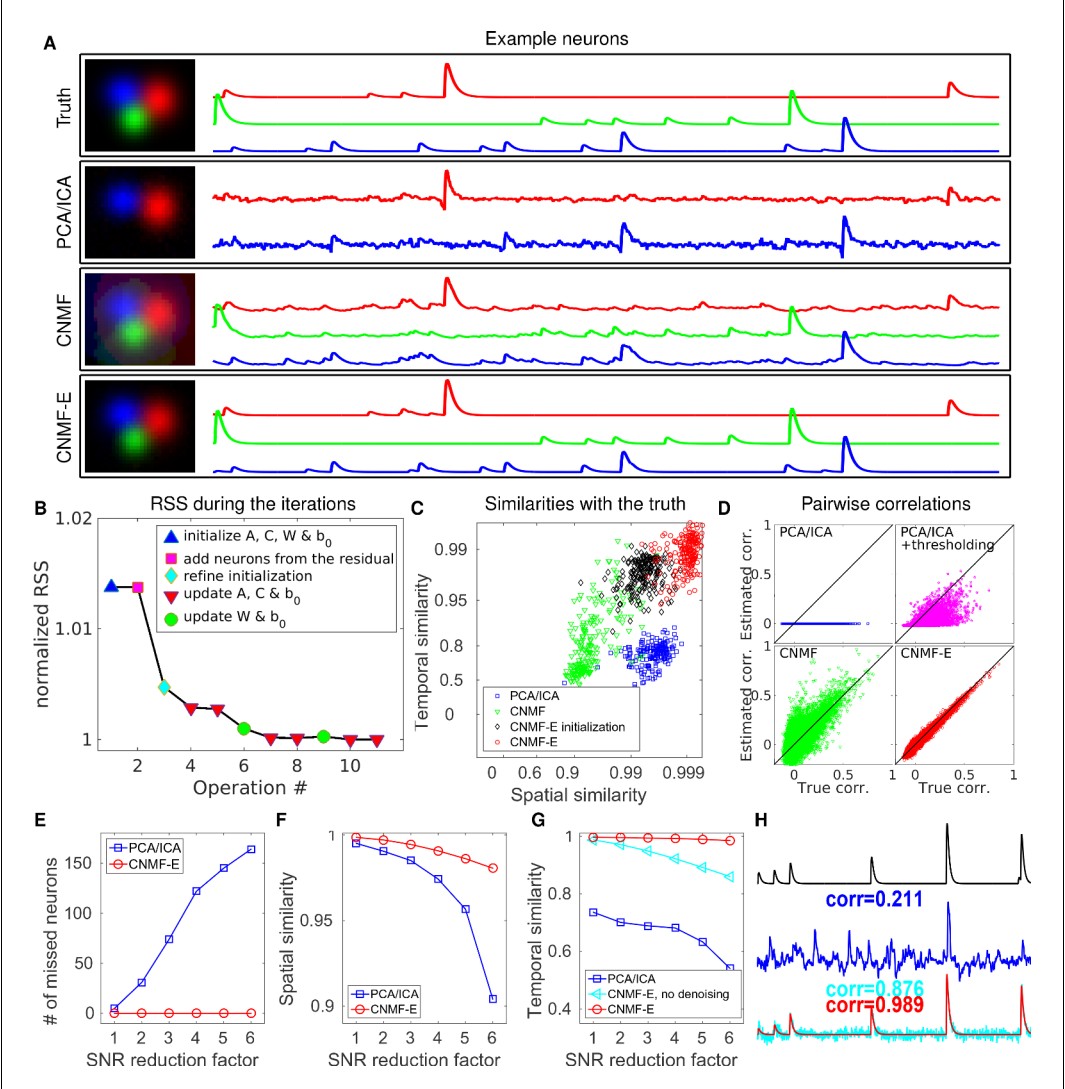

**Figure 4.** CNMF-E outperforms PCA/ICA analysis in extracting individual neurons' activity from simulated data and is robust to low SNR. (**A**) The results of PCA/ICA, CNMF, and CNMF-E in recovering the spatial footprints and temporal traces of three example neurons. The trace colors match the neuron colors shown in the left. (**B**) The intermediate residual sum of squares (RSS) values (normalized by the final RSS value), during the CNMF-E model fitting. The 'refine initialization' step refers to the modification of the initialization results in the case of high temporal correlation (details in Materials and methods). (**C**) The spatial and the temporal cosine similarities between the ground truth and the neurons detected using different methods. (**D**) The pairwise correlations between the calcium activity traces extracted using different methods. (**E–G**) The performances of PCA/ICA and CNMF-E under different noise levels: the number of missed neurons (**E**), and the spatial (**F**) and temporal (**G**) cosine similarities between the extracted components and the ground truth. (**H**) The calcium traces of one example neuron: the ground truth (black), the PCA/ICA trace (blue), the CNMF-E trace (red) and the CNMF-E trace without being denoised (cyan). The similarity values shown in the figure are computed as the cosine similarity between each trace and the ground truth (black). Two videos showing the demixing results of the simulated data can be found in *Video 4* (SNR reduction factor = 1) and *Video 5* (SNR reduction factor = 6).

DOI: https://doi.org/10.7554/eLife.28728.009

updating the model variables $A, C$, and $B$. The first two iterations were performed automatically; we included manual interventions (e.g. merging/deleting components) before the last iteration, leading to improved source extraction results (see *Video 10* for details on the manual merge and delete interventions performed here). In this example, we detected 24 CNMF-E components and 24 PCA/ICA components. The contours of these inferred neurons are shown in *Figure 8A*. In total we have 20 components detected by both methods (shown in the first three rows of *Figure 8B+C*); each method detected extra components that are not detected by the other (the last rows of *Figure 8B*

+C). Once again, the PCA/ICA filters contain many negative pixels in an effort to reduce spatial overlaps; see components 3 and 5 in *Figure 8A–C*, for example. All traces of the inferred neurons are shown in *Figure 8D+E*. We can see that the CNMF-E traces have much lower noise level and cleaner neural signals in both high and low SNR settings. Conversely, the calcium traces of the three extra neurons identified by PCA/ICA show noisy signals that are unlikely to be neural responses.

## Application to footshock responses in the bed nucleus of the stria terminalis (BNST)

Identifying neurons and extracting their temporal activity is typically just the first step in the analysis of calcium imaging data; downstream analyses rely heavily on the quality of this initial source extraction. We showed above that, compared to PCA/ICA, CNMF-E is better at extracting activity dynamics, especially in regimes where neuronal activities are correlated (c.f. *Figure 4D*). Using in vivo electrophysiological recordings, we previously showed that neurons in the bed nucleus of the stria terminalis (BNST) show strong responses to unpredictable footshock stimuli (*Jennings et al., 2013*). We therefore measured calcium dynamics in CaMKII-expressing neurons that were transfected with the calcium indicator GCaMP6s in the BNST and analyzed the synchro-

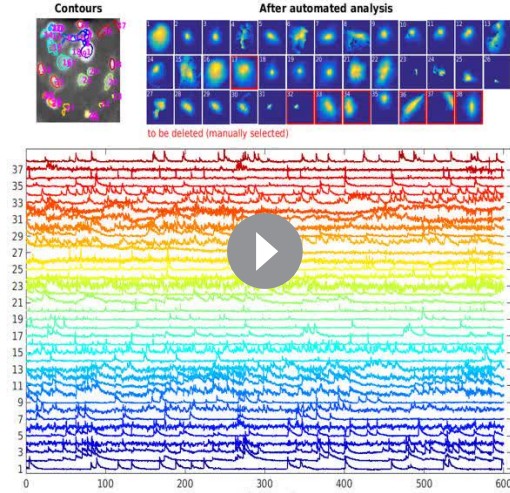

**Video 10.** Extracted spatial and temporal components of CNMF-E at different stages (ventral hippocampal dataset). After initializing components, we ran matrix updates and interventions in automatic mode, resulting in 32 components in total. In the next iteration, we manually deleted 6 components and automatically merged neurons as well. In the last iterations, 4 neurons were merged into 2 neurons with manual verifications. The correlation image in the top left panel is computed from the background-subtracted data in the final step. MP4

DOI: https://doi.org/10.7554/eLife.28728.020

nous activity of multiple neurons in response to unpredictable footshock stimuli. We chose 12 example neurons that were detected by both CNMF-E and PCA/ICA methods and show their spatial and temporal components in *Figure 9A–C*. The activity around the onset of the repeated stimuli are aligned and shown as pseudo-colored images in panel D. The median responses of CNMF-E neurons display prominent responses to the footshock stimuli compared with the resting state before stimuli onset. In comparison, the activity dynamics extracted by PCA/ICA have relatively low SNR, making it more challenging to reliably extract footshock responses. Panel E summarizes the results of panel D; we see that CNMF-E outputs significantly more easily detectable responses than does PCA/ICA. This is an example in which downstream analyses of calcium imaging data can significantly benefit from the improvements in the accuracy of source extraction offered by CNMF-E. (sheintuch2017-tracking recently presented another such example, showing that more neurons can be tracked across multiple days using CNMF-E outputs, compared to PCA/ICA.)

## Conclusion

Microendoscopic calcium imaging offers unique advantages and has quickly become a critical method for recording large neural populations during unrestrained behavior. However, previous methods fail to adequately remove background contaminations when demixing single neuron activity from the raw data. Since strong background signals are largely inescapable in the context of one-photon imaging, insufficient removal of the background could yield problematic conclusions in downstream analysis. This has presented a severe and well-known bottleneck in the field. We have delivered a solution for this critical problem, building on the constrained nonnegative matrix factorization framework introduced in *Pnevmatikakis et al., 2016* but significantly extending it in order to more accurately and robustly remove these contaminating background components.

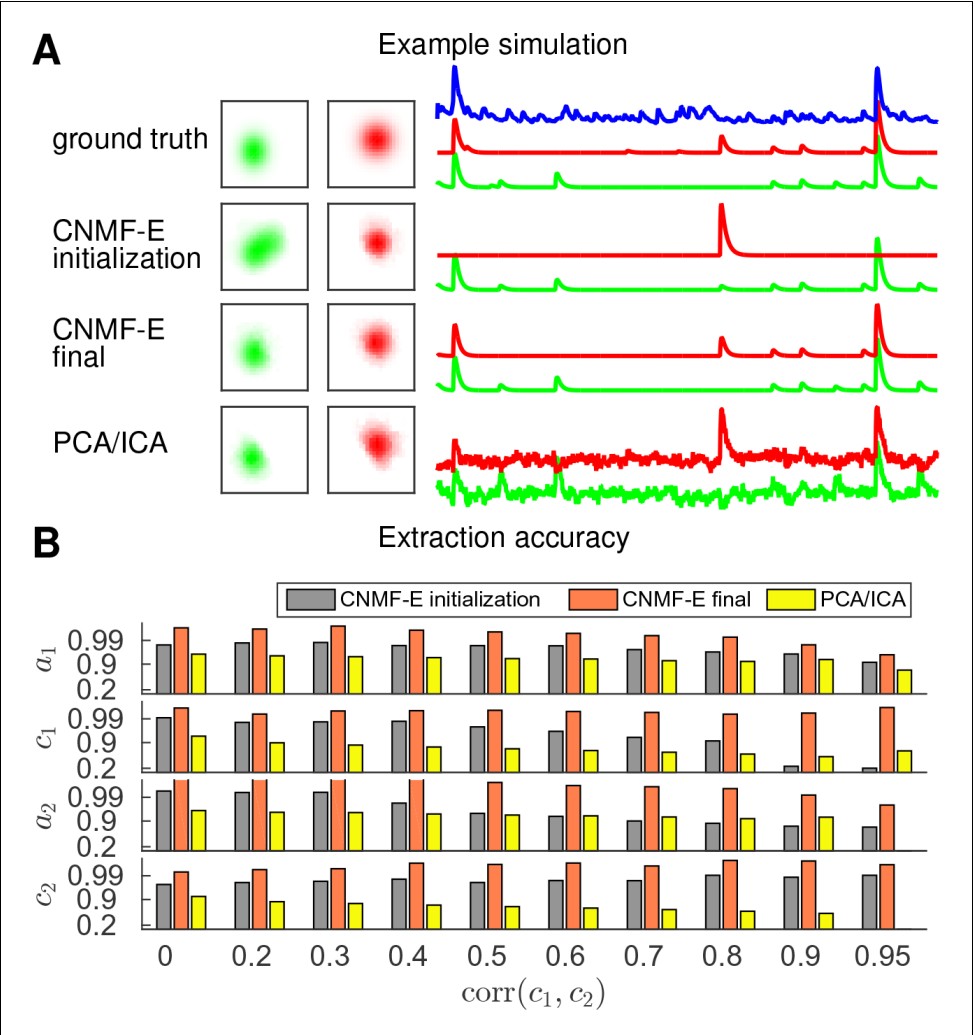

**Figure 5.** CNMF-E is able to demix neurons with high temporal correlations. (**A**) An example simulation from the experiments summarized in panel (**B**), where $\mathrm{corr}(c_1, c_2)$ is 0.9: green and red traces correspond to the corresponding neuronal shapes in the left panels. The blue trace is the mean background fluorescence fluctuation over the whole FOV. (**B**) The extraction accuracy of the spatial ($a_1$ and $a_2$) and the temporal ($c_1$ and $c_2$) components of two close-by neurons, computed via the cosine similarity between the ground truth and the extraction results.
DOI: https://doi.org/10.7554/eLife.28728.012

The proposed CNMF-E algorithm can be used in either automatic or semi-automatic mode, and leads to significant improvements in the accuracy of source extraction compared with previous methods. In addition, CNMF-E requires very few parameters to be specified, and these parameters are easily interpretable and can be selected within a broad range. We demonstrated the power of CNMF-E using data from a wide diversity of brain areas (subcortical, cortical, and deep brain areas), SNR regimes, calcium indicators, neuron sizes and densities, and hardware setups. Among all these examples (and many others not shown here), CNMF-E performs well and improves significantly on the standard PCA/ICA approach. Considering that source extraction is typically just the first step in calcium imaging data analysis pipelines (*Mohammed et al., 2016*), these improvements should in turn lead to more stable and interpretable results from downstream analyses. Further applications of the CNMF-E approach appear in (*Cameron et al., 2016*; *Donahue and Kreitzer, 2017*; *Jimenez et al., 2016*; *Jimenez et al., 2018*; *Klaus et al., 2017*; *Lin et al., 2017*; *Murugan et al., 2016*; *Murugan et al., 2017*; *Rodriguez-Romaguera et al., 2017*; *Tombaz et al., 2016*; *Ung et al.,*

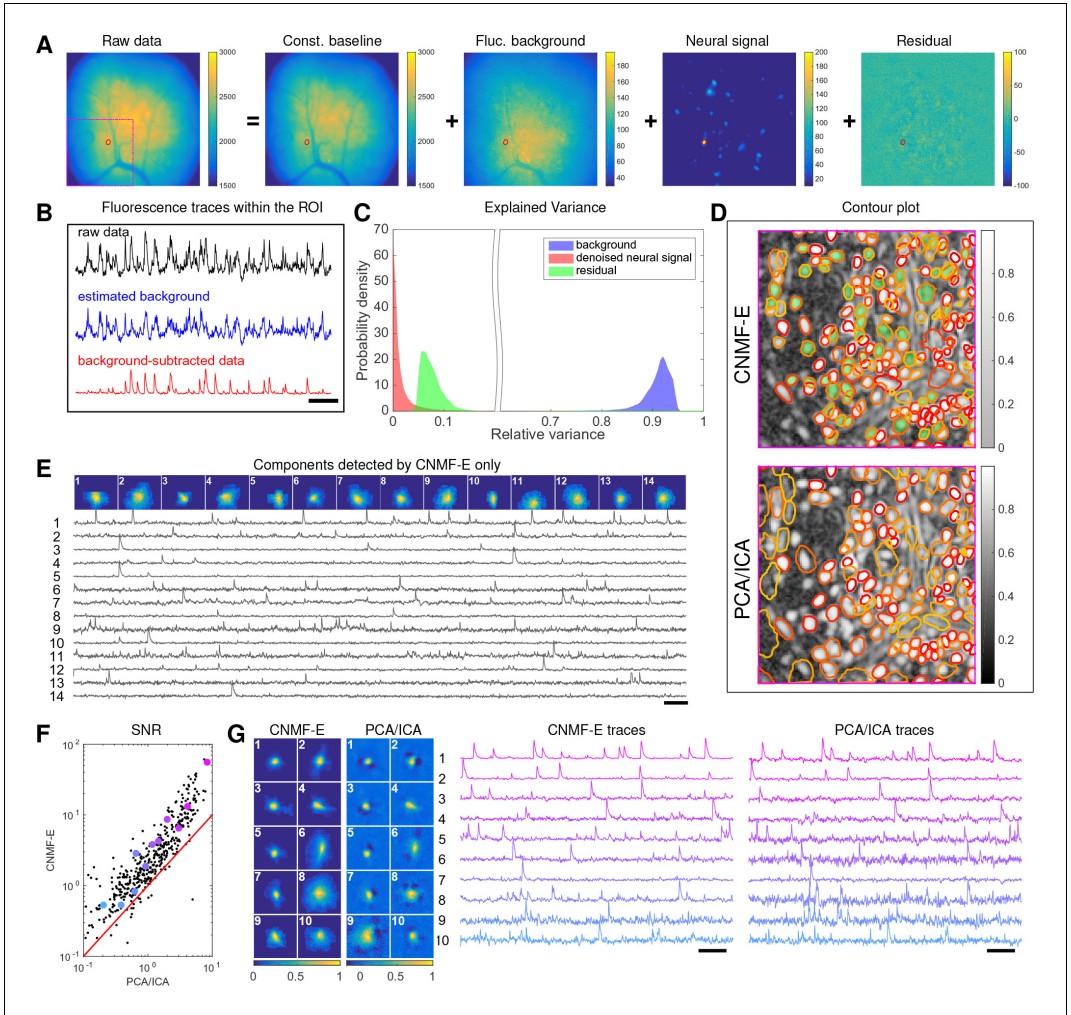

**Figure 6.** Neurons expressing GCaMP6f recorded in vivo in mouse dorsal striatum area. (**A**) An example frame of the raw data and its four components decomposed by CNMF-E. (**B**) The mean fluorescence traces of the raw data (black), the estimated background activity (blue), and the background-subtracted data (red) within the segmented area (red) in (**A**). The variance of the black trace is about 2x the variance of the blue trace and 4x the variance of the red trace. (**C**) The distributions of the variance explained by different components over all pixels; note that estimated background signals dominate the total variance of the signal. (**D**) The contour plot of all neurons detected by CNMF-E and PCA/ICA superimposed on the correlation image. Green areas represent the components that are only detected by CNMF-E. The components are sorted in decreasing order based on their SNRs (from red to yellow). (**E**) The spatial and temporal components of 14 example neurons that are only detected by CNMF-E. These neurons all correspond to green areas in (**D**). (**F**) The signal-to-noise ratios (SNRs) of all neurons detected by both methods. Colors match the example traces shown in (**G**), which shows the spatial and temporal components of 10 example neurons detected by both methods. Scalebar: 10 s. See **Video 6** for the demixing results.

DOI: https://doi.org/10.7554/eLife.28728.013

*2017*; *Yu et al., 2017*; *Mackevicius et al., 2017*; *Madangopal et al., 2017*; *Roberts et al., 2017*; *Ryan et al., 2017*; *Roberts et al., 2017*; *Sheintuch et al., 2017*).

We have released our MATLAB implementation of CNMF-E as open-source software (https://github.com/zhoupc/CNMF_E (*Zhou, 2017a*)). A Python implementation has also been incorporated into the CalmAn toolbox (*Giovannucci et al., 2017b*). We welcome additions or suggestions for modifications of the code, and hope that the large and growing microendoscopic imaging community finds CNMF-E to be a helpful tool in furthering neuroscience research.

## Materials and methods

### Algorithm for solving problem (P-S)

In problem (P-S), $b_0$ is unconstrained and can be updated in closed form: $\hat{b}_0 = \frac{1}{T}(\tilde{Y} - A \cdot \hat{C} - \hat{B}^f) \cdot 1$. By plugging this update into problem (P-S), we get a reduced problem

$$\underset{A}{\text{minimize}} \quad \|\tilde{Y} - A \cdot \tilde{C}\|_F^2 \tag{P-S'}$$
$$\text{subject to} \quad A \geq 0, \; A \text{ is local and sparse,}$$

where $\tilde{Y} = Y - \hat{B}^f - \frac{1}{T}Y11^T$ and $\tilde{C} = \hat{C} - \frac{1}{T}\hat{C}11^T$. We approach this problem using a version of "hierarchical alternating least squares' (HALS; *Cichocki et al., 2007*), a standard algorithm for nonnegative matrix factorization. (*Friedrich et al., 2017b*) modified the fastHALS algorithm (*Cichocki and Phan, 2009*) to estimate the nonnegative spatial components $A,b$ and the nonnegative temporal activity $C,f$ in the CNMF model $Y = A \cdot C + bf^T + E$ by including sparsity and localization constraints. We solve a problem similar to the subproblem solved in *Friedrich et al. (2017b)*:

$$\underset{A}{\text{minimize}} \quad \|\tilde{Y} - A \cdot \tilde{C}\|_F^2 \tag{P-S'}$$
$$\text{subject to} \quad A \geq 0, \; A \text{ is local and sparse,}$$

where $P_k$ denotes the the spatial patch constraining the nonzero pixels of the $k$-th neuron and restricts the candidate spatial support of neuron $k$. This regularization reduces the number of free parameters in $A$, leading to speed and accuracy improvements. The spatial patches can be determined using a mildly dilated version of the support of the previous estimate of $A$ (*Pnevmatikakis et al., 2016*; *Friedrich et al., 2017a*).

### Algorithms for solving problem (P-T)

In problem (P-T), the model variable $C \in \mathbb{R}^{K \times T}$ could be very large, making the direct solution of (P-T) computationally expensive. Unlike problem (P-S), the problem (P-T) cannot be readily parallelized because the constraints $G^{(i)}c_i \geq 0$ couple the entries within each row of C, and the residual term couples entries across columns. Here, we follow the block coordinate-descent approach used in (*Pnevmatikakis et al., 2016*) and propose an algorithm that sequentially updates each $c_i$ and $b_0$. For each neuron, we start with a simple unconstrained estimate of $c_i$, denoted as $\hat{y}_i$, that minimizes the residual of the spatiotemporal data matrix while fixing other neurons' spatiotemporal activity and the baseline term $b_0$,

$$\hat{y}_i = \underset{c_i \in \mathbb{R}^T}{argmin} \|Y - \hat{A}_{\backslash i} \cdot \hat{C}_{\backslash i} - \hat{a}_i c_i - \hat{b}_0 \cdot I^T - \hat{B}^f\|_F^2 = \hat{c}_i + \frac{\hat{a}_i^T \cdot Y_{\text{res}}}{\hat{a}_i^T \hat{a}_i}, \tag{9}$$

where $Y_{\text{res}} = Y - \hat{A}\hat{C} - \hat{b}_0 I^T - B^f$ represents the residual given the current estimate of the model variables. Due to its unconstrained nature, $\hat{y}_i$ is a noisy estimate of $c_i$, plus a constant baseline resulting from inaccurate estimation of $b_0$. Given $\hat{y}_i$, various deconvolution algorithms can be applied to obtain the denoised trace $\hat{c}_i$ and deconvolved signal $\hat{s}_i$(*Vogelstein et al., 2009*; *Pnevmatikakis et al., 2013*; *Deneux et al., 2016*; *Friedrich et al., 2017b*; *Jewell and Witten, 2017*); in CNMF-E, we use the OASIS algorithm from (*Friedrich et al., 2017b*). (Note that the estimation of $c_i$ is not dependent on accurate estimation of $b_0$, because the algorithm for estimating $c_i$ will also automatically estimate the baseline term in $\hat{y}_i$.) After the $c_i$'s are updated,

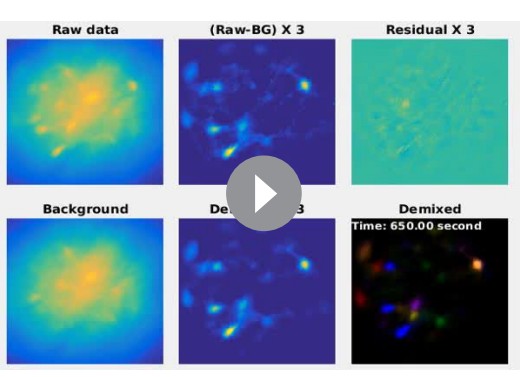

**Video 11.** The results of CNMF-E in demixing BNST data. Conventions as in *Video 6*. MP4
DOI: https://doi.org/10.7554/eLife.28728.022

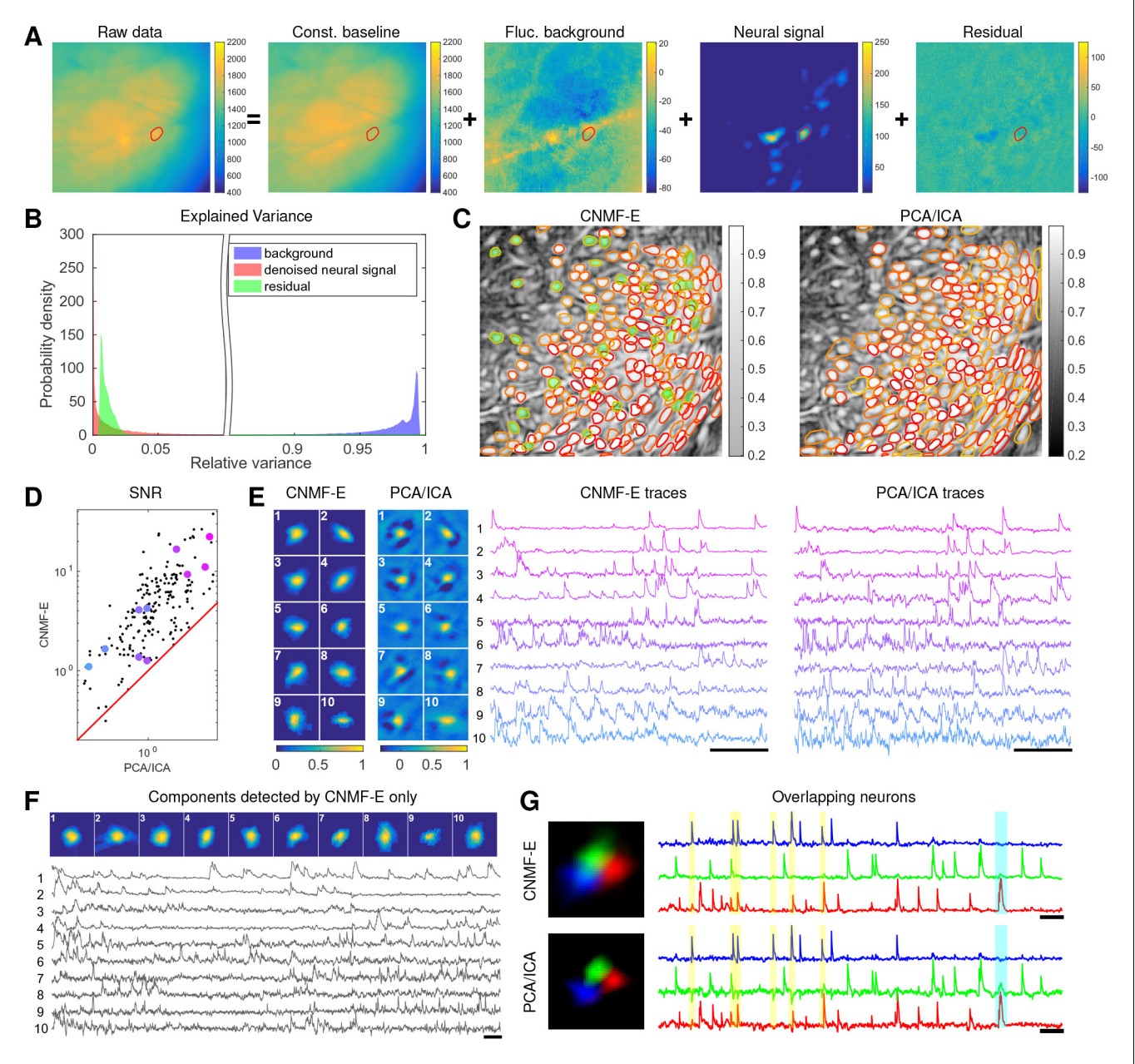

**Figure 7.** Neurons expressing GCaMP6s recorded in vivo in mouse prefrontal cortex. (**A–F**) follow similar conventions as in the corresponding panels of *Figure 6*. (**G**) Three example neurons that are close to each other and detected by both methods. Yellow shaded areas highlight the negative 'spikes' correlated with nearby activity, and the cyan shaded area highlights one crosstalk between nearby neurons. Scalebar: 20 s. See *Video 7* for the demixing results and *Video 8* for the comparision of CNMF-E and PCA/ICA in the zoomed-in area of (**G**).
DOI: https://doi.org/10.7554/eLife.28728.015

we update $b_0$ using the closed-form expression $\hat{b}_0 = \frac{1}{T}(\tilde{Y} - \hat{A} \cdot \hat{C} - \hat{B}^f) \cdot 1$.

## Estimating background by solving problem (P-B)

Next we discuss our algorithm for estimating the spatiotemporal background signal by solving problem (P-B) as a linear regression problem given $\hat{A}$ and $\hat{C}$. Since $B^f \cdot 1 = 0$, we can easily estimate the constant baselines for each pixel as

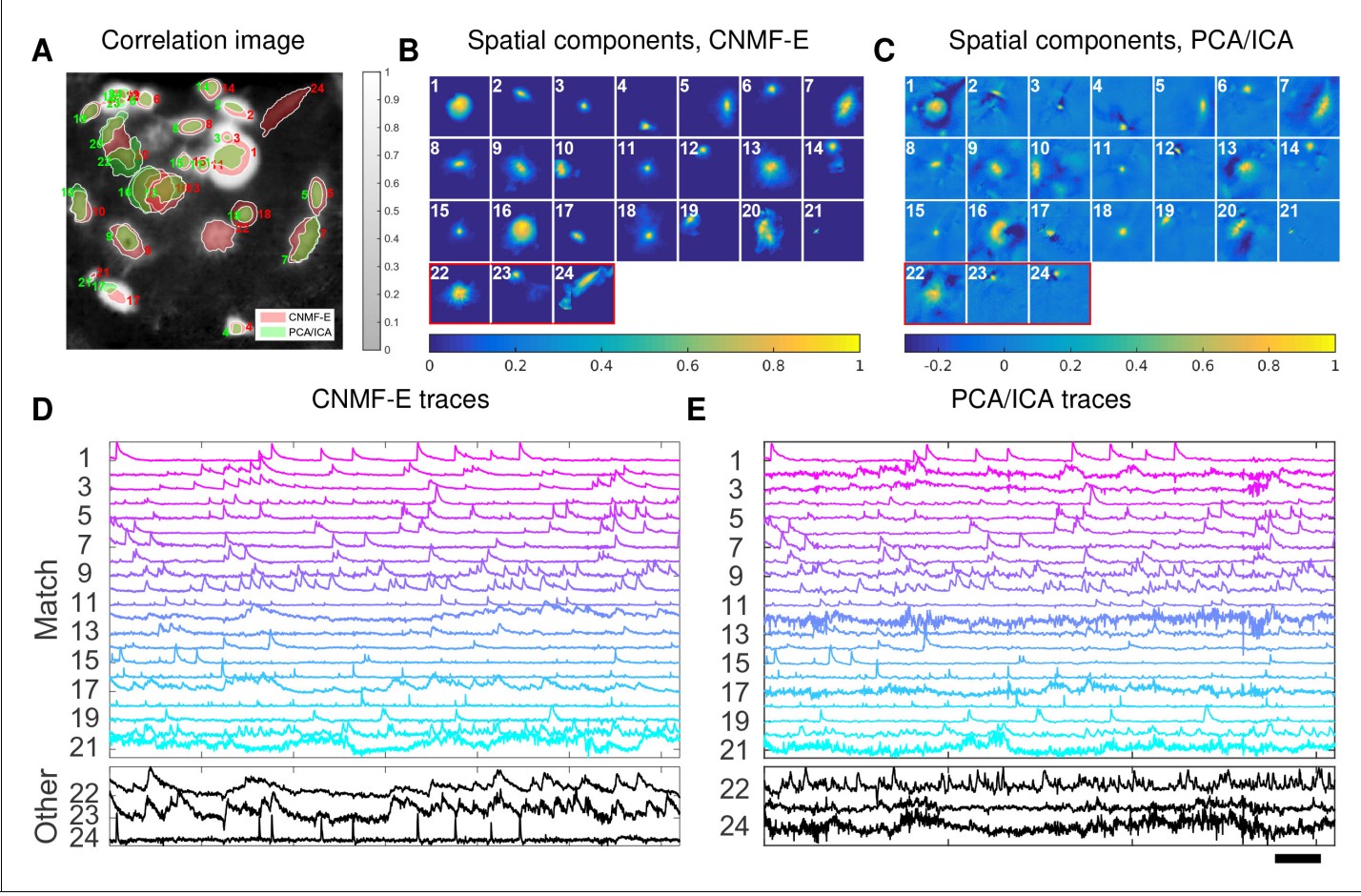

**Figure 8.** Neurons expressing GCaMP6f recorded in vivo in mouse ventral hippocampus. (A) Contours of all neurons detected by CNMF-E (red) and PCA/ICA method (green). The grayscale image is the local correlation image of the background-subtracted video data, with background estimated using CNMF-E. (B) Spatial components of all neurons detected by CNMF-E. The neurons in the first three rows are also detected by PCA/ICA, while the neurons in the last row are only detected by CNMF-E. (C) Spatial components of all neurons detected by PCA/ICA; similar to (B), the neurons in the first three rows are also detected by CNMF-E and the neurons in the last row are only detected by PCA/ICA method. (D) Temporal traces of all detected components in (B). 'Match' indicates neurons in top three rows in panel (B); 'Other' indicates neurons in the fourth row. (E) Temporal traces of all components in (C). Scalebars: 20 seconds. See *Video 9* for demixing results.

DOI: https://doi.org/10.7554/eLife.28728.018

$$\hat{b}_0 = \frac{1}{T}(Y - \hat{A} \cdot \hat{C}) \cdot 1. \tag{10}$$

Next we replace the $b_0$ in (P-B) with this estimate and rewrite (P-B) as

$$\begin{aligned} \underset{W}{\text{minimize}} \quad & \|X - W \cdot X\|_F^2, \\ \text{subject to} \quad & W_{ij} = 0 \text{ if } \operatorname{dist}(\boldsymbol{x}_i, \boldsymbol{x}_j) \notin [l_n, l_n + 1), \end{aligned} \tag{P-W}$$

where $X = Y - \hat{A} \cdot \hat{C} - \hat{b}_0 1^T$. Given the optimized $\hat{W}$, our estimation of the fluctuating background is $\hat{B}^f = \hat{W}X$. The new optimization problem (P-W) can be readily parallelized into $d$ linear regression problems for each pixel separately. By estimating all row columns of $W_{i,:}$, we are able to obtain the whole background signal as

$$\hat{B} = \hat{W}X + \hat{b}_0 1^T. \tag{11}$$

In some cases, $X$ might include large residuals from the inaccurate estimation of the neurons'

spatiotemporal activity $AC$, for example, missing neurons in the estimation. These residuals act as outliers and distort the estimation of $\hat{B}^f$ and $b_0$. To overcome this problem, we use robust least squares regression (RLSR) via hard thresholding to avoid contaminations from the outliers (*Bhatia et al., 2015*). Before solving the problem (P-W), we compute $B^- = \hat{W}(Y - \hat{A} \cdot \hat{C} - \hat{b}_0 1^T)$ (the current estimate of the fluctuating background) and then apply a simple clipping preprocessing step to $X$:

$$X_{it}^{clipped} = \begin{cases} B_{it}^- & \text{if } X_{it} \geq B_{it}^- + \zeta \cdot \sigma_i \\ X_{it} & \text{otherwise} \end{cases}.$$ (12)

Then we update the regression estimate using $X^{clipped}$ instead of $X$, and iterate. Here, $\sigma_i$ is the standard deviation of the noise at $x_i$ and its value can be estimated using the power spectral density (PSD) method (*Pnevmatikakis et al., 2016*). As for the first iteration of the model fitting, we set each $B_{it}^- = \frac{1}{|\Omega_i|}\sum_{j \in \Omega_i} \tilde{X}_{jt}$ as the mean of the $\tilde{X}_{jt}$ for all $j \in \Omega_i$. The thresholding coefficient $\zeta$ can be specified by users, although we have found a fixed default works well across the datasets used here. This preprocessing removes most calcium transients by replacing those frames with the previously estimated background only. As a result, it increases the robustness to inaccurate estimation of $AC$, and in turn leads to a better extraction of $AC$ in the following iterations.

## Initialization of model variables

Since problem (P-All) is not convex in all of its variables, a good initialization of model variables is crucial for fast convergence and accurate extraction of all neurons' spatiotemporal activity. Previous methods assume the background component is relatively weak, allowing us to initialize $\hat{A}$ and $\hat{C}$ while ignoring the background or simply initializing it with a constant baseline over time. However, the noisy background in microendoscopic data fluctuates more strongly than the neural signals (c.f. *Figure 6C* and *Figure 7B*), which makes previous methods less valid for the initialization of CNMF-E.

Here, we design a new algorithm to initialize $\hat{A}$ and $\hat{C}$ without estimating $\hat{B}$. The whole procedure is illustrated in *Figure 10* and described in Algorithm 1. The key aim of our algorithm is to exploit the relative spatial smoothness in the background compared to the single neuronal signals visible in the focal plane. Thus, we can use spatial filtering to reduce the background in order to estimate single neurons' temporal activity, and then initialize each neuron's spatial footprint given these temporal traces. Once we have initialized $\hat{A}$ and $\hat{C}$, it is straightforward to initialize the constant baseline $b_0$ and the fluctuating background $B^f$ by solving problem (P-B).

### Spatially filtering the data

We first filter the raw video data with a customized image kernel (*Figure 10A*). The kernel is generated from a Gaussian filter

$$h(x) = \exp\left(-\frac{\|x\|^2}{2(l/4)^2}\right).$$ (13)

Here, we use $h(x)$ to approximate a cell body; the factor of $1/4$ in the Gaussian width is chosen to match a Gaussian shape to a cell of width $l$. Instead of using $h(x)$ as the filtering kernel directly, we subtract its spatial mean (computed over a region of width equal to $l$) and filter the raw data with $\tilde{h}(x) = h(x) - \bar{h}(x)$. The filtered data is denoted as $Z \in \mathbb{R}^{d \times T}$ (*Figure 10B*). This spatial filtering step helps accomplish two goals: (1) reducing the background $B$, so that $Z$ is dominated by neural signals (albeit somewhat spatially distorted) in the focal plane (see *Figure 10B* as an example); (2) performing a template matching to detect cell bodies similar to the Gaussian kernel. Consequently, $Z$ has large values near the center of each cell body. (However, note that we can not simply e.g. apply CNMF to $Z$, because the spatial components in a factorization of the matrix $Z$ will typically no longer be nonnegative, and therefore NMF-based approaches can not be applied directly.) More importantly, the calcium traces near the neuron center in the filtered data preserve the calcium activity of the corresponding neurons because the filtering step results in a weighted average of cellular signals surrounding each pixel (*Figure 10B*). Thus, the fluorescence traces in pixels close to neuron centers in $Z$ can be used for initializing the neurons'

temporal activity directly. These pixels are defined as seed pixels. We next propose a quantitative method to rank all potential seed pixels.

## Ranking seed pixels

A seed pixel $x$ should have two main features: first, $Z(x)$, which is the filtered trace at pixel $x$, should have high peak-to-noise ratio (PNR) because it encodes the calcium concentration $c_i$ of one neuron; second, a seed pixel should have high temporal correlations with its neighboring pixels (e.g. 4 nearest neighbors) because they share the same $c_i$. We computed two metrics for each of these two features:

$$P(\boldsymbol{x}) = \frac{\max_t(Z(\boldsymbol{x}, t))}{\sigma(\boldsymbol{x})}, \ L(\boldsymbol{x}) = \frac{1}{4} \sum_{\mathrm{dist}(\boldsymbol{x}, \boldsymbol{x}')=1} \mathrm{corr}\Big(Z(\boldsymbol{x}), \ Z(\boldsymbol{x}')\Big). \tag{14}$$

Recall that $\sigma(x)$ is the standard deviation of the noise at pixel $x$; the function $\mathbf{corr}()$ refers to Pearson correlation here. In our implementation, we usually threshold $Z(x)$ by $3\sigma(x)$ before computing $L(x)$ to reduce the influence of the background residuals, noise, and spikes from nearby neurons.

Most pixels can be ignored when selecting seed pixels because their local correlations or PNR values are too small. To avoid unnecessary searches of the pixels, we set thresholds for both $P(x)$ and $L(x)$, and only pick pixels larger than the thresholds $P_{\min}$ and $L_{\min}$. It is empirically useful to combine both metrics for screening seed pixels. For example, high PNR values could result from large noise, but these pixels usually have small $L(x)$ because the noise is not shared with neighboring pixels. On the other hand, insufficient removal of background during the spatial filtering leads to high $L(x)$, but the corresponding $P(x)$ are usually small because most background fluctuations have been removed. So we create another matrix $R(x) = P(x) \cdot L(x)$ that computes the pixelwise product of $P(x)$ and $L(x)$. We rank all $R(x)$ in a descending order and choose the pixel $x^*$ with the largest $R(x)$ for initialization.

---

**Algorithm 1. Initialize model variables $A$ and $C$ given the raw data**

---

**Require:** data $Y \in \mathbb{R}^{d \times T}$, neuron size $l$, the minimum local correlation $L_{min}$ and the minimum PNR $P_{min}$ for selecting seed pixels.

1:  $\mathrm{h} \leftarrow$ a truncated 2D Gaussian kernel of width$\sigma_\mathrm{x} = \sigma_\mathrm{y} = \frac{1}{4}$; $\mathrm{h} \in \mathbb{R}^{l \times l}$ ▷ 2D Gaussian kernel
2:  $\tilde{h} \leftarrow h - \bar{h}$; $\tilde{h} \in \mathbb{R}^{l \times l}$ ▷ mean − centered kernel for spatial filtering
3:  $Z \leftarrow \mathrm{conv}(Y, h)$; $Z \in \mathbb{R}^{d \times T}$ ▷ spatially filter the raw data
4:  $L \leftarrow$ local cross − correlation image of the filtered data Z; $L \in \mathbb{R}^d$
5:  $P \leftarrow$ PNR image of the filtered data Z; $P \in \mathbb{R}^d$
6:  $k \leftarrow 0$ ▷ neuron number
7:  **while** True **do**
8: **if** $L(\boldsymbol{x}) \leq L_{min}$ *or* $P(\boldsymbol{x}) \leq P_{min}$ for all pixel $\boldsymbol{x}$ **then**
9: break;
10: **else**
11: $k \leftarrow k + 1$
12: $\hat{a}_k \leftarrow \boldsymbol{0}$; $\boldsymbol{a} \in \mathbb{R}^d$
13: $\boldsymbol{x}^* \leftarrow \mathrm{argmax}_{\boldsymbol{x}}(L(\boldsymbol{x}) \cdot P(\boldsymbol{x}))$ ▷ select a seed pixel
14: $\Omega_k \leftarrow \{\boldsymbol{x} | \boldsymbol{x}$ is in the square box of length $(2l+1)$ surrounding pixel $\boldsymbol{x}^*\}$ ▷ crop a small box near $\boldsymbol{x}^*$
15: $\boldsymbol{r}(\boldsymbol{x}) \leftarrow \mathrm{corr}(Z(\boldsymbol{x}, :), Z(\boldsymbol{x}^*, :))$ for all $\boldsymbol{x} \in \Omega_k$; $\boldsymbol{r} \in \mathbb{R}^{|\Omega_k|}$
16: $\boldsymbol{y}_{BG} \leftarrow \dfrac{\sum_{\{\boldsymbol{x} | \boldsymbol{r}(\boldsymbol{x}) \leq 0.3\}} Y(\boldsymbol{x}, :)}{\sum_{\{\boldsymbol{x} | \boldsymbol{r}(\boldsymbol{x}) \leq 0.3\}} 1}$; $\boldsymbol{y}_{BG} \in \mathbf{R}^T$ ▷ estimate the background signal
17: $\hat{\boldsymbol{c}}_k \leftarrow \dfrac{\sum_{\{\boldsymbol{x} | \boldsymbol{x}(\boldsymbol{x}) \geq 0.7\}} Z(\boldsymbol{x}, :)}{\sum_{\{\boldsymbol{x} | \boldsymbol{r}(\boldsymbol{x}) \geq 0.7\}} 1}$; $\hat{\boldsymbol{c}}_k \in \mathbf{R}^T$ ▷ estimate neural signal
18: $\hat{\boldsymbol{a}}_k(\Omega_k), \hat{\boldsymbol{b}}^{(f)}, \hat{\boldsymbol{b}}^{(0)} \leftarrow \mathrm{argmin}_{\boldsymbol{a}, \boldsymbol{b}^{(f)}, \boldsymbol{b}^{(0)}} \|Y_{\Omega_k} - (\boldsymbol{a} \cdot \hat{\boldsymbol{c}}_k^T + \boldsymbol{b}^{(f)} \cdot \boldsymbol{y}_{BG}^T + \boldsymbol{b}^{(0)} \cdot \boldsymbol{1}^T)\|_F^2$
19: $\hat{\boldsymbol{a}}_k \leftarrow \max(0, \hat{\boldsymbol{a}}_k)$ ▷ the spatial component of the k − th neuron
20: $Y \leftarrow Y - \hat{\boldsymbol{a}}_k \cdot \hat{\boldsymbol{c}}_k^T$ ▷ peel away the neuron's spatiotemporal activity
21: update L($\boldsymbol{x}$) and P($\boldsymbol{x}$) locally given the new Y
22: $A \leftarrow [\hat{\boldsymbol{a}}_1, \hat{\boldsymbol{a}}_2, \cdots, \hat{\boldsymbol{a}}_k]$
23: $C \leftarrow [\hat{\boldsymbol{c}}_1, \hat{\boldsymbol{c}}_2, \cdots, \hat{\boldsymbol{c}}_k]^T$
24: **return** $A, C$

---

## Greedy initialization

Our initialization method greedily initializes neurons one by one. Every time we initialize a neuron, we will remove its initialized spatiotemporal activity from the raw video data and initialize the next neuron from the residual. For the same neuron, there are several seed pixels that could be used to

initialize it. But once the neuron has been initialized from any of these seed pixels (and the spatio-temporal residual matrix has been updated by peeling away the corresponding activity), the remaining seed pixels related to this neuron have lowered PNR and local correlation. This helps avoid the duplicate initialization of the same neuron. Also, $P(x)$ and $L(x)$ have to be updated after each neuron is initialized, but since only a small area near the initialized neuron is affected, we can update these quantities locally to reduce the computational cost. This procedure is repeated until the specified number of neurons have been initialized or no more candidate seed pixels exist.

This initialization algorithm can greedily initialize the required number of neurons, but the sub-problem of estimating $\hat{a}_i$ given $\hat{c}_i$ still has to deal with the large background activity in the residual matrix. We developed a simple method to remove this background and accurately initialize neuron shapes, described next. We first crop a $(2l + 1) \times (2l + 1)$ square centered at $x^*$ in the field of view (*Figure 10A–E*). Then we compute the temporal correlation between the filtered traces of pixel $x^*$ and all other pixels in the patch (*Figure 10D*). We choose those pixels with small temporal correlations (e.g. 0.3) as the neighboring pixels that are outside of the neuron (the green contour in *Figure 10D*). Next, we estimate the background fluctuations as the median values of these pixels for each frame in the raw data (*Figure 10E*). We also select pixels that are within the neuron by selecting correlation coefficients larger than 0.7, then $\hat{c}_i$ is refined by computing the mean filtered traces of these pixels (*Figure 10E*). Finally, we regress the raw fluorescence signal in each pixel onto three sources: the neuron signal (*Figure 10E*), the local background fluctuation (*Figure 10F*), and a constant baseline. Our initial estimate of $\hat{a}_i$ is given by the regression weights onto $\hat{c}_i$ in *Figure 10F*.

## Modifications for high temporal or spatial correlation

The above procedure works well in most experimental datasets as long as neurons are not highly spatially overlapped and temporally correlated. However, in a few extreme cases, this initialization may lead to bad local minima. We found that two practical modifications can lead to improved results.

### High temporal correlation, low spatial overlaps

The greedy initialization procedure assumes that closeby neurons are not highly correlated. If this assumption fails, CNMF-E will first merge nearby neurons into one component for explaining the shared fluctuations, and then the following initialized components will only capture the residual signals of each neuron. Our solution to this issue relies on our accurate background removal procedure, after which we simply re-estimate each neural trace $c_i$ as a weighted fluorescence trace of the background-subtracted video $(Y - \hat{B}^f - \hat{b}_0 1^T)$,

$$\hat{c}_i = \frac{\tilde{a}^T \cdot (Y - \hat{B}^f - \hat{b}_0 1^T)}{\tilde{a}^T \cdot \tilde{a}}, \tag{15}$$

where $\tilde{a}_i$ only selects pixels with large weights by thresholding the estimated $\hat{a}_i$ with $\max(\hat{a}_i)/2$ (this reduces the contributions from smaller neighboring neurons). This strategy improves the extraction of individual neurons' traces in the high correlation scenarios and the spatial footprints can be corrected in the following step of updating $\hat{A}$. *Figure 4B* and *Figure 5* illustrate this procedure.

### High spatial overlaps, low temporal correlation

CNMF-E may initialize components with shared temporal traces because they have highly overlapping areas. We solve this problem by de-correlating their traces (following a similar approach in [*Pnevmatikakis et al., 2016*]). We start by assuming that neurons with high spatial overlap do not fire spikes within the same frame. If so, only the inferred spiking trace with the largest value is kept and the rest will be set to 0. Then we initialize each $c_i$ given these thresholded spiking traces and the corresponding AR coefficients.

## Interventions

We use iterative matrix updates to estimate model variables in CNMF-E. This strategy gives us the flexibility of integrating prior information on neuron morphology and temporal activity during the model fitting. The resulting interventions (which can in principle be performed either automatically or under manual control) can in turn lead to faster convergence and more accurate source

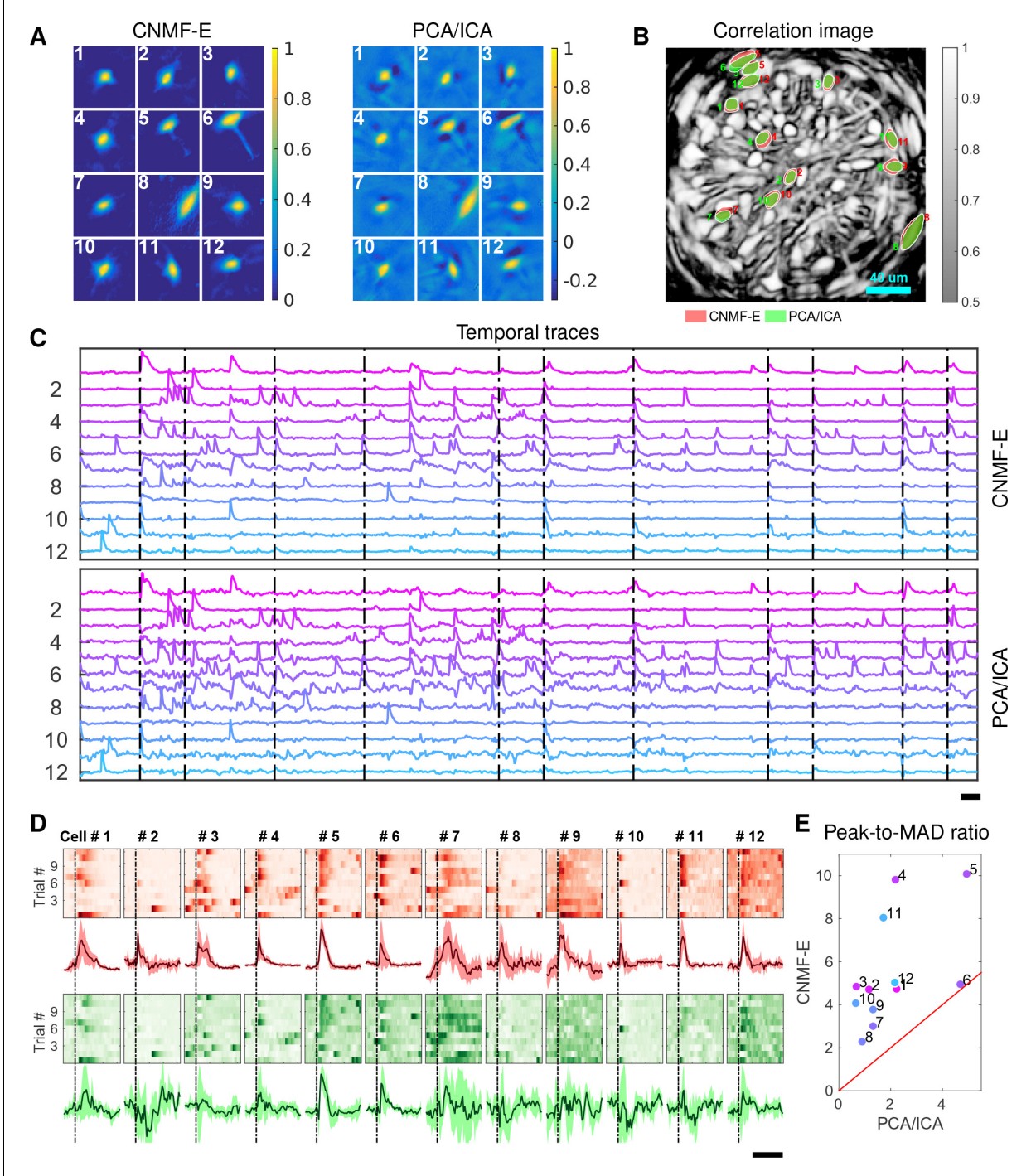

**Figure 9.** Neurons extracted by CNMF-E show more reproducible responses to footshock stimuli, with larger signal sizes relative to the across-trial variability, compared to PCA/ICA. (**A–C**) Spatial components (**A**), spatial locations (**B**) and temporal components (**C**) of 12 example neurons detected by both CNMF-E and PCA/ICA. (**D**) Calcium responses of all example neurons to footshock stimuli. Colormaps show trial-by-trial responses of each neuron, extracted by CNMF-E (top, red) and PCA/ICA (bottom, green), aligned to the footshock time. The solid lines are medians of neural responses over 11 trials and the shaded areas correpond to median $\pm 1$ median absolute deviation (MAD). Dashed lines indicate the shock timings. (**E**) Scatter plot of peak-to-MAD ratios for all response curves in (**D**). For each neuron, Peak is corrected by subtracting the mean activity within 4 s prior to stimulus onset and MAD is computed as the mean MAD values over all timebins shown in (**D**). The red line shows $y = x$. Scalebars: 10 s. See ***Video 11*** for demixing results.

DOI: https://doi.org/10.7554/eLife.28728.021

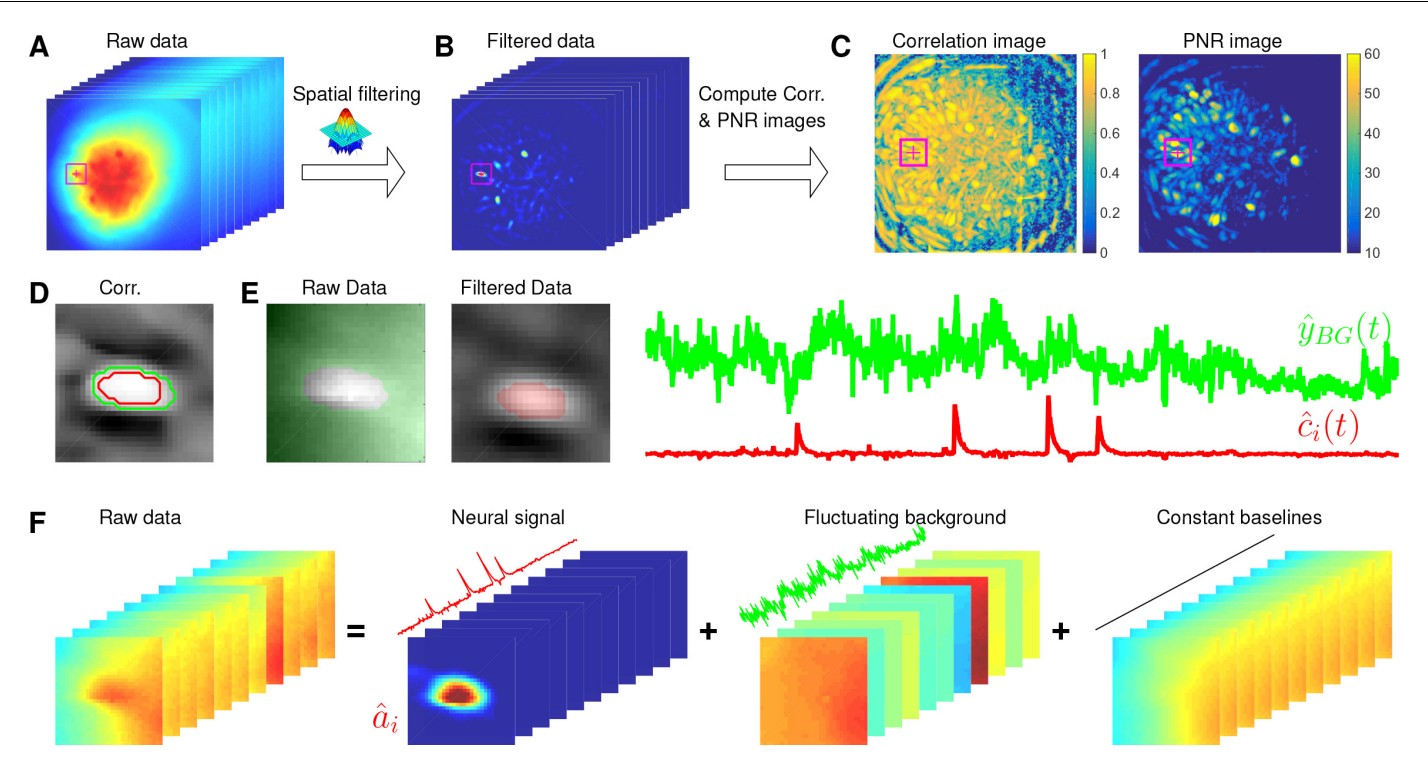

**Figure 10.** Illustration of the initialization procedure. (**A**) Raw video data and the kernel for filtering the video data. (**B**) The spatially high-pass filtered data. (**C**) The local correlation image and the peak-to-noise ratio (PNR) image calculated from the filtered data in (**B**). (**D**) The temporal correlation coefficients between the filtered traces (**B**) of the selected seed pixel (the red cross) and all other pixels in the cropped area as shown in (**A–C**). The red and green contour correspond to correlation coefficients equal to 0.7 and 0.3, respectively. (**E**) The estimated background fluctuation $y_{BG}(t)$ (green) and the initialized temporal trace $\hat{c}_i(t)$ of the neuron (red). $y_{BG}(t)$ is computed as the median of the raw fluorescence traces of all pixels (green area) outside of the green contour shown in (**D**) and $\hat{c}_i(t)$ is computed as the mean of the filtered fluorescence traces of all pixels inside the red contour. (**F**) The decomposition of the raw video data within the cropped area. Each component is a rank-1 matrix and the related temporal traces are estimated in (**E**). The spatial components are estimated by regressing the raw video data against these three traces. See *Video 3* for an illustration of the initialization procedure.

DOI: https://doi.org/10.7554/eLife.28728.023

extraction. We integrate 5 interventions in our CNMF-E implementation. Following these interventions, we usually run one more iteration of matrix updates.

## Merge existing components

When a single neuron is split mistakenly into multiple components, a merge step is necessary to rejoin these components. If we can find all split components, we can superimpose all their spatiotemporal activities and run rank-1 NMF to obtain the spatial and temporal activity of the merged neuron. We automatically merge components for which the spatial and temporal components are correlated above certain thresholds. Our code also provides methods to manually specify neurons to be merged based on human judgment.

## Split extracted components

When highly correlated neurons are mistakenly merged into one component, we need to use spatial information to split into multiple components according to neurons' morphology. Our current implementation of component splitting requires users to manually draw ROIs for splitting the spatial footprint of the extracted component. Automatic methods for ROI segmentation (*Apthorpe et al., 2016*; *Pachitariu et al., 2013*) could be added as an alternative in future implementations.

**Table 1.** Variables used in the CNMF-E model and algorithm. $\mathbb{R}$: real numbers; $\mathbb{R}_+$: positive real numbers; $\mathbb{N}$: natural numbers; $\mathbb{N}_+$: positive integers.

| Name | Description | Domain |
|------|-------------|--------|
| $d$ | number of pixels | $\mathbb{N}_+$ |
| $T$ | number of frames | $\mathbb{N}_+$ |
| $K$ | number of neurons | $\mathbb{N}$ |
| $Y$ | motion corrected video data | $\mathbb{R}_+^{d \times T}$ |
| $A$ | spatial footprints of all neurons | $\mathbb{R}_+^{d \times K}$ |
| $C$ | temporal activities of all neurons | $\mathbb{R}_+^{K \times T}$ |
| $B$ | background activity | $\mathbb{R}_+^{d \times T}$ |
| $E$ | observation noise | $\mathbb{R}^{d \times T}$ |
| $W$ | weight matrix to reconstruct $B$ using neighboring pixels | $\mathbb{R}^{d \times d}$ |
| $b_0$ | constant baseline for all pixels | $\mathbb{R}_+^{d}$ |
| $x_i$ | spatial location of the $i$th pixel | $\mathbb{N}^2$ |
| $\sigma_i$ | standard deviation of the noise at pixel $x_i$ | $\mathbb{R}_+$ |

DOI: https://doi.org/10.7554/eLife.28728.006

## Remove false positives

Some extracted components have spatial shapes that do not correspond to real neurons or temporal traces that do not correspond to neural activity. These components might explain some neural signals or background activity mistakenly. Our source extraction can benefit from the removal of these false positives. This can be done by manually examining all extracted components, or in principle automatically by training a classifier for detecting real neurons. The current implementation relies on visual inspection to exclude false positives. We also rank neurons based on their SNRs and set a cut-off to discard all extracted components that fail to meet this cutoff. As with the splitting step, removing false positives could also potentially use automated ROI detection algorithms in the future. See *Video 10* for an example involving manual merge and delete operations.

## Pick undetected neurons from the residual

If all neural signals and background are accurately estimated, the residual of the CNMF-E model $Y_{\text{res}} = Y - \hat{A}\hat{C} - \hat{B}$ should be relatively spatially and temporally uncorrelated. However, the initialization might miss some neurons due to large background fluctuations and/or high neuron density. After we estimate the background $\hat{B}$ and extract a majority of the neurons, those missed neurons have prominent fluorescent signals left in the residual. To select these undetected neurons from the residual $Y_{\text{res}}$, we use the same algorithm as for initializing neurons from the raw video data, but typically now the task is easier because the background has been removed.

## Post-process the spatial footprints

Each single neuron has localized spatial shapes and including this prior into the model fitting of CNMF-E, as suggested in (*Pnevmatikakis et al., 2016*), leads to better extraction of spatial footprints. In the model fitting step, we constrain $A$ to be sparse and spatially localized. These constraints do give us compact neuron shapes in most cases, but in some cases there are still some visually abnormal components detected. We include a heuristic automated post-processing step after each iteration of updating spatial shapes (P-S). For each extracted neuron $A(:, k)$, we first convert it to a 2D image and perform morphological opening to remove isolated pixels resulting from noise (*Haralick et al., 1987*). Next we label all connected components in the image and create a mask to select the largest component. All pixels outside of the mask in $A(:, i)$ are set to be. This post-processing induces compact neuron shapes by removing extra pixels and helps avoid mistakenly explaining the fluorescence signals of the other neurons.

## Further algorithmic details

The simplest pipeline for running CNMF-E includes the following steps:

1. Initialize $\hat{A}, \hat{C}$ using the proposed initialization procedure.
2. Solve problem (P-B) for updates of $\hat{b}_0$ and $\hat{B}^f$.
3. Iteratively solve problem (P-S) and (P-T) to update $\hat{A}, \sim \hat{C}$ and $b_0$.
4. If desired, apply interventions to intermediate results.
5. Repeat steps 2, 3, and 4 until the inferred components are stable.

In practice, the estimation of the background $B$ (step 2) often does not vary greatly from iteration to iteration and so this step usually can be run with fewer iterations to save time. In practice, we also use spatial and temporal decimation for improved speed, following (*Friedrich et al., 2017a*). We first run the pipeline on decimated data to get good initializations, then we up-sample the results $\hat{A}, \hat{C}$ to the original resolution and run one iteration of steps (2-3) on the raw data. This strategy improves on processing the raw data directly because downsampling increases the signal-to-noise ratio and eliminates many false positives.

Step 4 provides a fast method for correcting abnormal components without redoing the whole analysis. (This is an important improvement over the PCA/ICA pipeline, where if users encounter poor estimated components it is necessary to repeat the whole analysis with new parameter values, which may not necessarily yield improved cell segmentations.) The interventions described here themselves can be independent tasks in calcium imaging analysis; with further work we expect many of these steps can be automated. In our interface for performing manual interventions, the most frequently used function is to remove false positives. Again, components can be rejected following visual inspection in PCA/ICA analysis, but the performance of CNMF-E can be improved with further iterations after removing false positives, while this is not currently an option for PCA/ICA.

We have also found a two-step initialization procedure useful for detecting neurons: we first start from relatively high thresholds of $P_{\min}$ and $L_{\min}$ to initialize neurons with large activity from the raw video data; then we estimate the background components by solving problem (P-B); finally we can pick undetected neurons from the residual using smaller thresholds. We can terminate the model iterations when the residual sum of squares (RSS) stabilizes (see *Figure 4B*), but this is seldom used in practice because computing the RSS is time-consuming. Instead we usually automatically stop the iterations after the number of detected neurons stabilizes. If manual interventions are performed, we typically run one last iteration of updating $B, A$ and $C$ sequentially to further refine the results.

## Parameter selection

*Table 2* shows 5 key parameters used in CNMF-E. All of these parameters have interpretable meaning and can be easily picked within a broad range. The parameter $l$ controls the size of the spatial filter in the initialization step and is chosen as the diameter of a typical neuron in the FOV. As long as $l$ is much smaller than local background sources, the filtered data can be used for detecting seed pixels and then initializing neural traces. The distance between each seed pixel and its selected neighbors $l_n$ has to be larger than the neuron size $l$ and smaller than the spatial range of local background sources; in practice, this range is fairly broad. We usually set $l_n$ as $2l$. To determine the thresholds $P_{\min}$ and $L_{\min}$, we first compute the correlation image and PNR image and then visually select very weak neurons from these two images. $P_{\min}$ and $L_{\min}$ are determined to ensure that CNMF-E is able to choose seed pixels from these weak neurons. Small $P_{\min}$ and $L_{\min}$ yield more false positive neurons, but they can be removed in the intervention step. Finally, in practice, our results are not sensitive to the selection of the outlier parameter $\zeta$, thus we frequently set it as 10.

## Complexity analysis

In step 1, the time cost is mainly determined by spatial filtering, resulting in $O(dT)$ time. As for the initialization of a single neuron given a seed pixel, it is only $(O(T))$. Considering the fact that the number of neurons is typically much smaller than the number of pixels in this data, the complexity for step one remains $O(dT)$. In step 2, the complexity of estimating $\hat{b}_0$ is $O(dT)$ and estimating $\hat{B}^f$ scales linearly with the number of pixels $d$. For each pixel, the computational complexity for estimating $W_{i,:}$ is $O(T)$. Thus, the computational complexity in updating the background component is $O(dT)$. In step 3, the computational complexities of solving problems (P-S) and (P-T) have been discussed in

**Table 2.** Optional user-specified parameters.

| Name | Description | Default values | Used in |
|---|---|---|---|
| $l$ | size of a typical neuron soma in the FOV | $30\mu m$ | Algorithm 1 |
| $l_n$ | the distance between each pixel and its neighbors | $60\mu m$ | Problem (P-B) |
| $P_{\min}$ | the minimum peak-to-noise ratio of seed pixels | 10 | Algorithm 1 |
| $L_{\min}$ | the minimum local correlation of seed pixels | 0.8 | Algorithm 1 |
| $\zeta$ | the ratio between the outlier threshold and the noise | 10 | Problem (P-B) |

DOI: https://doi.org/10.7554/eLife.28728.024

previous literature (*Pnevmatikakis et al., 2016*) and they scale linearly with pixel number $d$ and time $T$, that is, $O(dT)$. For the interventions, the one with the largest computational cost is picking undetected neurons from the residual, which is the same as the initialization step. Therefore, the computational cost for step 4 is $O(dT)$. To summarize, the complexity for running CNMF-E is $O(dT)$, that is, the method scales linearly with both the number of pixels and the total recording time.

## Implementations

Our MATLAB implementation supports running CNMF-E in three different modes that are optimized for different datasets: single-mode, patch-mode and multi-batch-mode.

Single-mode is a naive implementation that loads data into memory and fits the model. It is fast for processing small datasets (<1 GB).

For larger datasets, many computers have insufficient RAM for loading all data into memory and storing intermediate results. Patch-mode CNMF-E divides the whole FOV into multiple small patches and maps data to the hard drive (*Giovannucci et al., 2017b*). The data within each patch are loaded only when we process that patch. This significantly reduces the memory consumption. More importantly, this mode allows running CNMF-E in parallel on multi-core CPUs, yielding a speed-up roughly proportional to the number of available cores.

Multi-batch mode builds on patch-mode and is optimized for even larger datasets, especially data collected over multiple sessions/days. This mode segments data into multiple batches temporally and assumes that the neuron footprints $A$ are shared across all batches. We process each batch using patch mode and perform partial weighted updates on $A$ given the traces $C$ obtained in each batch.

All modes also include a logging system for keeping track of manual interventions and intermediate operations.

The Python implementation is similar; see *Giovannucci et al., 2017b*) for full details.

## Running time

To provide a sense of the running time of the different steps of the algorithm, we timed the code on the simulation data shown in *Figure 4*. This dataset is $253 \times 316$ pixels $\times 2000$ frames. The analyses were performed on a desktop with Intel Xeon CPU E5-2650 v4 @2.20 GHz and 128 GB RAM running Ubuntu 16.04. We used a parallel implementation for performing the CNMF-E analysis, with patch size $64 \times 64$ pixels, using up to 12 cores. PCA/ICA took $\sim 211$ seconds to converge, using 250 PCs and 220 ICs. CNMF-E spent 55 s for initialization, 1 s for merging and deleting components, 110 s for the first round of the background estimation and 40 s in the following updates, 8 s for picking neurons from the residual, and 10 s per iteration for updating spatial ($A$) and temporal ($C$) components, resulting in a total of 258 s.

Finally, *Table 3* shows the running time of processing the four experimental datasets.

## Simulation experiments

### Details of the simulated experiment of *Figure 2*

The field of view was $256 \times 256$, with 1000 frames. We simulated 50 neurons whose shapes were simulated as spherical 2-D Gaussian. The neuron centers were drawn uniformly from the whole FOV and the Gaussian widths $\sigma_x$ and $\sigma_y$ for each neuron was also randomly drawn from $\mathcal{N}\left(\frac{l}{4}, \left(\frac{1}{10}\frac{l}{4}\right)^2\right)$, where $l = 12$ pixels. Spikes were simulated from a Bernoulli process with probability of spiking per timebin

**Table 3.** Running time (sec) for processing the 4 experimental datasets.

| Dataset | Striatum | PFC | Hippocampus | BNST |
|---|---|---|---|---|
| Size ($x \times y \times t$) | $256 \times 256 \times 6000$ | $175 \times 184 \times 9000$ | $175 \times 184 \times 9000$ | $175 \times 184 \times 9000$ |
| (# PCs, # ICs) | (2000, 700) | (275, 250) | (100, 50) | (200, 150) |
| PFC/ICA | 986 | 181 | 174 | 52 |
| CNMF-E | 726 | 221 | 225 | 435 |

DOI: https://doi.org/10.7554/eLife.28728.025

0.01 and then convolved with a temporal kernel $g(t) = \exp(-t/\tau_d) - \exp(-t/\tau_r)$, with fall time $\tau_d = 6$ timebin and rise time $\tau_r = 1$ timebin. We simulated the spatial footprints of local backgrounds as 2-D Gaussian as well, but the mean Gaussian width is 5 times larger than the neurons' widths. As for the spatial footprint of the blood vessel in *Figure 2A*, we simulated a cubic function and then convolved it with a 2-D Gaussian (Gaussian width=3pixel). We use a random walk model to simulate the temporal fluctuations of local background and blood vessel. For the data used in *Figure 2A–H*, there were 23 local background sources; for *Figure 2I*, we varied the number of background sources.

We used the raw data to estimate the background in CNMF-E without subtracting the neural signals $\hat{A}\hat{C}$ in problem (P-B). We set $l_n = 15$ pixels and left the remaining parameters at their default values. The plain NMF was performed using the built-in MATLAB function nnmf, which utilizes random initialization.

## Details of the simulated experiment of *Figure 3*, *Figure 4* and *Figure 5*

We used the same simulation settings for both *Figure 3* and *Figure 4*. The field of view was $253 \times 316$ and the number of frames was 2000. We simulated 200 neurons using the same method as the simulation in *Figure 2*, but for the background we used the spatiotemporal activity of the background extracted using CNMF-E from real experimental data (data not shown). The noise level $\Sigma$ was also estimated from the data. When we varied the SNR in *Figure 4D–G*, we multiplied $\Sigma$ with an SNR reduction factor.

We set $l = 12$ pixels to create the spatial filtering kernel. As for the thresholds used for determining seed pixels, we varied them for different SNR settings by visually checking the corresponding local correlation images and PNR images. The selected values were $L_{\min} = [0.9, 0.8, 0.8, 0.8, 0.6, 0.6]$ and $P_{\min} = [15, 10, 10, 8, 6, 6]$ for different SNR reduction factors $[1, 2, 3, 4, 5, 6]$. For PCA/ICA analysis, we set the number of PCs and ICs as 600 and 300, respectively.

The simulation in *Figure 5* only includes two neurons (as seen in *Figure 3E*) using the same simulation parameters. We replaced their temporal traces $c_1$ and $c_2$ with $(1 - \rho)c_1 + \rho c_3$ and $(1 - \rho)c_2 + \rho c_3$, where $\rho$ is tuned to generate different correlation levels ($\gamma$), and $c_3$ is simulated in the same way as $c_1$ and $c_2$. We also added a new background source whose temporal profile is $c_3$ to increase the neuron-background correlation as $\rho$ increases. CNMF-E was run as in *Figure 4*. We used 20 PCs and ICs for PCA/ICA.

### In vivo microendoscopic imaging and data analysis

For all experimental data used in this work, we ran both CNMF-E and PCA/ICA. For CNMF-E, we chose parameters so that we initialized about 10–20% extra components, which were then merged or deleted (some automatically, some under manual supervision) to obtain the final estimates. Exact parameter settings are given for each dataset below. For PCA/ICA, the number of ICs were selected to be slightly larger than our extracted components in CNMF-E (as we found this led to the best results for this algorithm), and the number of PCs was selected to capture over 90% of the signal variance. The weight of temporal information in spatiotemporal ICA was set as 0.1. After obtaining PCA/ICA filters, we again manually removed components that were clearly not neurons based on neuron morphology.

We computed the SNR of extracted cellular traces to quantitatively compare the performances of two approaches. For each cellular trace $y$, we first computed its denoised trace $c$ using the selected deconvolution algorithm (here, it is thresholded OASIS); then the SNR of $y$ is

$$SNR = \frac{\|c\|_2^2}{\|y-c\|_2^2}. \tag{16}$$

For PCA/ICA results, the calcium signal $y$ of each IC is the output of its corresponding spatial filter, while for CNMF-E results, it is the trace before applying temporal deconvolution, that is, $\hat{y}_i$ in *Equation (9)*. All the data can be freely accessed online (*Zhou et al., 2017*).

## Dorsal striatum data

Expression of the genetically encoded calcium indicator GCaMP6f in neurons was achieved using a recombinant adeno-associated virus (AAV) encoding the GCaMP6f protein under transcriptional control of the synapsin promoter (AAV-Syn-GCaMP6f). This viral vector was packaged (Serotype 1) and stored in undiluted aliquots at a working concentration of >1012 genomic copies per ml at $-80°C$ until intracranial injection. 500 $\mu$l of AAV1-Syn-GCaMP6f was injected unilaterally into dorsal striatum (0.6 mm anterior to Bregma, 2.2mm lateral to Bregma, 2.5mm ventral to the surface of the brain). 1 week post-injection, a 1mm gradient index of refraction (GRIN) lens was implanted into dorsal striatum $\sim 300\mu$m above the center of the viral injection. Three weeks after the implantation, the GRIN lens was reversibly coupled to a miniature one-photon microscope with an integrated 475nm LED (Inscopix). Using nVistaHD Acquisition software, images were acquired at 30 frames per second with the LED transmitting $\sim 0.1$ to $0.2$ mW of light while the mouse was freely moving in an open-field arena. Images were down sampled to 10Hz and processed into TIFFs using Mosaic software. All experimental manipulations were performed in accordance with protocols approved by the Harvard Standing Committee on Animal Care following guidelines described in the US NIH Guide for the Care and Use of Laboratory Animals.

The parameters used in running CNMF-E were: $l = 13$ pixels, $l_n = 18$ pixels, $L_{\min} = 0.7$, and $P_{\min} = 7.728$ components were initialized from the raw data in the first pass before subtracting the background, and then additional components were initialized in a second pass. Highly correlated nearby components were merged and false positives were removed using the automated approach described above. In the end, we obtained 692 components.

## Prefrontal cortex data

Cortical neurons were targeted by administering two microinjections of 300 ul of AAV-DJ-CamkIIa-GCaMP6s (titer: $5.3 \times 1012$, 1:6 dilution, UNC vector core) into the prefrontal cortex (PFC) (coordinates relative to bregma; injection 1: +1.5 mm AP, 0.6 mm ML, $-2.4$ ml DV; injection 2: +2.15 AP, 0.43 mm ML, $-2.4$ mm DV) of an adult male wild type (WT) mice. Immediately following the virus injection procedure, a 1 mm diameter GRIN lens implanted 300 um above the injection site (coordinates relative to bregma: +1.87 mm AP, 0.5 mm ML, $-2.1$ ml DV). After sufficient time had been allowed for the virus to express and the tissue to clear underneath the lens (3 weeks), a baseplate was secured to the skull to interface the implanted GRIN lens with a miniature, integrated microscope (nVista, 473 nm excitation LED, Inscopix) and subsequently permit the visualization of Ca2 +signals from the PFC of a freely behaving mouse. The activity of PFC neurons were recorded at 15 Hz over a 10 min period (nVista HD Acquisition Software, Inscopix) while the test subject freely explored an empty novel chamber. Acquired data was spatially down sampled by a factor of 2, motion corrected, and temporally down sampled to 15 Hz (Mosaic Analysis Software, Inscopix). All procedures were approved by the University of North Carolina Institutional Animal Care and Use Committee (UNC IACUC).

The parameters used in running CNMF-E were: $l = 13$ pixels, $l_n = 18$ pixels, $L_{\min} = 0.9$, and $P_{\min} = 15$. There were 169 components initialized in the first pass and we obtained 225 components after running the whole CNMF-E pipeline.

## Ventral hippocampus data

The calcium indicator GCaMP6f was expressed in ventral hippocampal-amygdala projecting neurons by injecting a retrograde canine adeno type 2-Cre virus (CAV2-Cre; from Larry Zweifel, University of Washington) into the basal amydala (coordinates relative to bregma: $-1.70$ AP, 3.00 mm ML, and $-4.25$ mm DV from brain tissue at site), and a Cre-dependent GCaMP6f adeno associated virus (AAV1-flex-Synapsin-GCaMP6f, UPenn vector core) into ventral CA1 of the hippocampus

(coordinates relative to bregma: $-3.16$ mm AP, $3.50$ mm ML, and $-3.50$ mm DV from brain tissue at site). A 0.5 mm diameter GRIN lens was then implanted over the vCA1 subregion and imaging began 3 weeks after surgery to allow for sufficient viral expression. Mice were then imaged with Inscopix miniaturized microscopes and nVistaHD Acquisition software as described above; images were acquired at 15 frames per second, while mice explored an anxiogenic Elevated Plus Maze arena. Videos were motion corrected and spatially downsampled using Mosaic software. All procedures were performed in accordance with protocols approved by the New York State Psychiatric Institutional Animal Care and Use Committee following guidelines described in the US NIH Guide for the Care and Use of Laboratory Animals.

The parameters used in running CNMF-E were: $l = 15$ pixels, $l_n = 30$ pixels, $\zeta = 10$, $L_{\min} = 0.9$, and $P_{\min} = 15$. We first temporally downsampled the data by 2. Then we applied CNMF-E to the downsampled data. There were 53 components initialized. After updating the background component, the algorithm detected six more neurons from the residual. We merged most of these components and deleted false positives. In the end, there were 24 components left. The intermediate results before and after each manual intervention are shown in *Video 10*.

## BNST data with footshock

Calcium indicator GCaMP6s was expressed within CaMKII-expressing neurons in the BNST by injecting the recombinant adeno-associated virus AAVdj-CaMKII-GCaMP6s (packaged at UNC Vector Core) into the anterior dorsal portion of BNST (coordinates relative to bregma: $0.10$ mm AP, $-0.95$ mm ML, $-4.30$ mm DV). A 0.6 mm diameter GRIN lens was implanted above the injection site within the BNST. As described above, images were acquired using a detachable miniature one-photon microscope and nVistaHD Acquisition Software (Inscopix). Images were acquired at 20 frames per second while the animal was freely moving inside a sound-attenuated chamber equipped with a house light and a white noise generator (Med Associates). Unpredictable foot shocks were delivered through metal bars in the floor as an aversive stimulus during a 10 min session. Each unpredictable foot shock was 0.75 mA in intensity and 500 ms in duration on a variable interval (VI-60). As described above, images were motion corrected, downsampled and processed into TIFFs using Mosaic Software. These procedures were conducted in adult C57BL/6J mice (Jackson Laboratories) and in accordance with the Guide for the Care and Use of Laboratory Animals, as adopted by the NIH, and with approval from the Institutional Animal Care and Use Committee of the University of North Carolina at Chapel Hill (UNC).

The parameters used in running CNMF-E were: $l = 15$ pixels, $l_n = 23$ pixels, $\zeta = 10$, $L_{\min} = 0.9$, and $P_{\min} = 15$. There were 149 components initialized and we detected 29 more components from the residual after estimating the background. there were 127 components left after running the whole pipeline.

## Code availability

All analyses were performed with custom-written MATLAB code. MATLAB implementations of the CNMF-E algorithm can be freely downloaded from https://github.com/zhoupc/CNMF_E (*Zhou, 2017a*). We also implemented CNMF-E as part of the Python package CaImAn (*Giovannucci et al., 2017b*), a computational analysis toolbox for large-scale calcium imaging and behavioral data (https://github.com/simonsfoundation/CaImAn [*Giovannucci et al., 2017a*]).

The scripts for generating all figures and the experimental data in this paper can be accessed from https://github.com/zhoupc/eLife_submission (*Zhou, 2017b*).

## Acknowledgements

We would like to thank CNMF-E users who received early access to our package and provided tremendously helpful feedback and suggestions, especially James Hyde, Jesse Wood, and Sean Piantadosi in Susanne Ahmari's lab in University of Pittsburgh, Andreas Klaus in Rui Costa's Lab in the Champalimaud Neurobiology of Action Laboratory, Suoqin Jin in Xiangmin Xu's lab at University of California - Irvine, Conor Heins at the National Institute of Drug Abuse, Chris Donahue in Anatol Kreitzer's lab at University of California - San Francisco, Xian Zhang in Bo Li's lab at Cold Spring Harbor Laboratory, Emily Mackevicius in Michale Fee's lab at Massachusetts Institute of Technology, Courtney Cameron and Malavika Murugan in Ilana Witten's lab at Princeton University, Pranav

Mamidanna in Jonathan Whitlock's lab at Norwegian University of Science and Technology, and Milekovic Tomislav in Gregoire Courtine's group at EPFL. We also thank Andreas Klaus for valuable comments on the manuscript.

## Additional information

### Funding

| Funder | Author |
| --- | --- |
| National Institute of Mental Health | Pengcheng Zhou<br>Jessica C Jimenez<br>Rene Hen<br>Mazen A Kheirbek<br>Robert E Kass |
| National Institute on Drug Abuse | Pengcheng Zhou<br>Jose Rodriguez-Romaguera<br>Garret D Stuber |
| Intelligence Advanced Research Projects Activity | Pengcheng Zhou<br>Liam Paninski |
| Defense Advanced Research Projects Agency | Liam Paninski |
| Army Research Office | Liam Paninski |
| National Institute of Biomedical Imaging and Bioengineering | Liam Paninski |
| Eunice Kennedy Shriver National Institute of Child Health and Human Development | Shanna L Resendez<br>Garret D Stuber |
| Howard Hughes Medical Institute | Jessica C Jimenez |
| National Institute on Aging | Jessica C Jimenez<br>Rene Hen |
| New York State Stem Cell Science | Jessica C Jimenez<br>Rene Hen |
| Hope for Depression Research Foundation | Jessica C Jimenez<br>Rene Hen |
| Canadian Institutes of Health Research | Shay Q Neufeld |
| Simons Foundation | Andrea Giovannucci<br>Johannes Friedrich<br>Eftychios A Pnevmatikakis<br>Garret D Stuber<br>Liam Paninski |
| International Mental Health Research Organization | Mazen A Kheirbek |
| National Institute of Neurological Disorders and Stroke | Bernardo L Sabatini |

The funders had no role in study design, data collection and interpretation, or the decision to submit the work for publication.

### Author contributions

Pengcheng Zhou, Conceptualization, Resources, Data curation, Software, Formal analysis, Validation, Investigation, Visualization, Methodology, Writing—original draft, Project administration, Writing—review and editing; Shanna L Resendez, Shay Q Neufeld, Resources, Data curation, Funding acquisition, Validation, Investigation, Writing—review and editing; Jose Rodriguez-Romaguera, Resources, Data curation, Validation, Investigation, Visualization, Writing—review and editing; Jessica C

Jimenez, Resources, Data curation, Funding acquisition, Validation, Investigation, Visualization, Writing—review and editing; Andrea Giovannucci, Johannes Friedrich, Eftychios A Pnevmatikakis, Software; Garret D Stuber, Rene Hen, Resources, Supervision, Funding acquisition; Mazen A Kheirbek, Resources, Supervision, Funding acquisition, Validation, Writing—review and editing; Bernardo L Sabatini, Robert E Kass, Resources, Supervision, Funding acquisition, Visualization, Writing—review and editing; Liam Paninski, Conceptualization, Resources, Supervision, Funding acquisition, Validation, Visualization, Methodology, Writing—original draft, Project administration, Writing—review and editing

### Author ORCIDs
Pengcheng Zhou (ID) http://orcid.org/0000-0003-1237-3931
Andrea Giovannucci (ID) http://orcid.org/0000-0002-7850-444X
Johannes Friedrich (ID) http://orcid.org/0000-0002-1321-5866
Eftychios A Pnevmatikakis (ID) https://orcid.org/0000-0003-1509-6394
Garret D Stuber (ID) http://orcid.org/0000-0003-1730-4855

### Ethics
Animal experimentation: These procedures were conducted in accordance with the Guide for the Care and Use of Laboratory Animals, as adopted by the NIH, and with approval from the Harvard Standing Committee on Animal Care (protocol number: IS00000571 ), or the University of North Carolina Institutional Animal Care and Use Committee (UNC IACUC, protocol number: 16-075.0), or the New York State Psychiatric Institutional Animal Care and Use Committee (protocol number: NYSPI-1412 ).

### Decision letter and Author response
Decision letter https://doi.org/10.7554/eLife.28728.030
Author response https://doi.org/10.7554/eLife.28728.031

## Additional files

### Supplementary files
• Transparent reporting form
DOI: https://doi.org/10.7554/eLife.28728.026

### Major datasets
The following dataset was generated:

| Author(s) | Year | Dataset title | Dataset URL | Database, license, and accessibility information |
|---|---|---|---|---|
| Zhou P, Resendez SL, Rodriguez-Romaguera J, Jimenez JC, Neufeld SQ, Giovannucci A, Friedrich J, Pnevmatikakis EA, Stuber GD, Hen R, Kheirbek MA, Sabatini BL, Kass RE, Paninski L | 2017 | Data from: Efficient and accurate extraction of in vivo calcium signals from microendoscopic video data | https://doi.org/10.5061/dryad.kr17k | Available at Dryad Digital Repository under a CC0 Public Domain Dedication |

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
