## [Decision Letter]

Thank you for submitting your article " Efficient and accurate extraction of in vivo calcium signals from microendoscopic video data" for consideration by *eLife*. Your article has been reviewed by three peer reviewers, and the evaluation has been overseen by David Van Essen as the Senior Editor and Reviewing Editor. One of the reviewers was Dr Timothy Holy (Reviewer 2); the other two have opted to remain anonymous.

The reviewers have discussed the reviews with one another and the Senior Editor has drafted this decision to help you prepare a revised submission. With regard to the issue raised with you by email as to whether the new algorithm's success is perhaps more of an accident of initialization than a reflection of good statistical design, the reviewers found your response to be thoughtful and helpful, but not entirely convincing. We therefore plan to ask the original reviewers to take another look at your revised manuscript before making a final decision.

Summary:

This is an excellent paper showing how the CNMF technique has been adapted for use with 1-P calcium imaging in endoscopic data. The paper is for the most part clearly written, and the simulations backed up by the in-vivo imaging data clearly demonstrates that this technique is by far superior to any other currently being used, including ICA/PCA or other variations. It builds on previous work from Pnevmatikakis and colleagues, differing primarily in the statistical model used to describe the background. It will likely be essential to use this technique to obtain the highest quality data, as background fluctuations and cross-talk between neurons can severely impact the data recorded.

However, some aspects of the analysis are difficult to judge, and there are a number of concerns and recommendations raised by the reviewers.

Essential revisions:

1) It is important to discuss carefully to what degree this new algorithm's success reflects more of an accident of initialization than good statistical design. Given that (a) SVD will always produce a lower reconstruction error than NMF for the same number of components, and that (b) having a high background might allow the neural components to "insulate" themselves from the constraints of nonnegativity by offsetting them from zero, it seems possible that the model also admits solutions that are better from the standpoint of the loss function yet worse from the standpoint of biological plausibility. As a consequence, it may only be the initialization causing the algorithm to land in a local (but far from global) minimum that is leading to the kinds of plausible solutions exhibited in the manuscript; the risk is that a more exhaustive optimization algorithm, applied to the same data and model, would prefer solutions that lack this plausibility.

2) The exposition of the CNMF-E algorithm in this paper could stand to be improved substantially. It is extremely hard (or, really, impossible) to make sense of the details of the proposed algorithm as presented in this paper! While the big picture of the algorithm is clear from a quick glance at Equation 8, the details are not clear from the many pages of text that follow.

3) There are concerns about how well the technique would work if the true firing correlations increase. Can the authors do a simulation where they increase pair-wise correlations between the deltaF traces systematically and see at what point segmentation breaks down or cross-talk removal artificially lowers the correlations.

4) Please provide more practical advice about how to implement the software. This method is very computationally intensive, and some direction needs to be given on how to run the software to allow a large number of movies to be analyzed in a reasonable amount of time. There are no benchmarks given (from my reading) on how long the analysis takes per minute of recording and how this can be optimized.

5) Please comment on success of being able to segment the same region over several recordings over days and match up neurons across days?

6) The authors should also present what happens when they iterate their model to full convergence (e.g. square root of the machine precision) and discuss the heuristics they use to choose when to terminate the iteration early.

7) How does the quality of the neuronal reconstruction compare if you just use CNMF on the spatially-filtered (and 0-truncated) image? If it performs similarly, it's not entirely clear that this more complex model represents much of an advance.

8) It is mentioned in subsection “in vivomicroendoscopic imaging and data analysis” that manual interventions were applied in the data analysis shown in this paper. This makes the comparisons shown in this paper unfair to competitors that are fully automated (since of course any method can be improved using manual intervention). Can the authors show results of CNMF-E without manual intervention? Also, how much manual investment (time, numbers of each type of decision, etc.) is necessary, and how much the manual intervention improves the result.

9) How does CNMF-E fare against CNMF when applied to two-photon data? i.e., when the additional flexibility of the model is perhaps not essential, does this extra flexibility degrade the performance in some fashion?

10) It is important to make the data and the scripts available for recreating the figures shown in this paper, for both the simulated and the real data. Otherwise it will be very hard for others to apply CNMF-E and get comparable results, given the many tuning parameters and semi-manual interventions involved.

11) What happens if one passes in the initialization described in subsection “Initialization of model variables” to the CNMF algorithm or to other competitors in the literature?

*Reviewer #1:*

This is an excellent paper showing how the CNMF technique has been adapted for use with 1-P calcium imaging in endoscopic data. The paper is very clearly written, and the simulations backed up by the in-vivo imaging data clearly demonstrates that this technique is by far superior to any other currently being used, including ICA/PCA or other variations. In fact, it will be essential to use this technique to obtain the highest quality data as background fluctuations and cross-talk between neurons can severely impact the data recorded.

These are my concerns:

1) I have some concerns about how well the technique would work if the true firing correlations increase. Can the authors do a simulation where they increase pair-wise correlations between the deltaF traces systematically and see at what point segmentation breaks down or cross-talk removal artificially lowers the correlations.

2 I felt that the authors could provide more practical advice about how to implement the software. This method is very computationally intensive, and some direction needs to be given on how to run the software to allow a large number of movies to be analyzed in a reasonable amount of time. There are no benchmarks given (from my reading) on how long the analysis takes per minute of recording and how this can be optimized.

3) Can the authors comment on success of being able to segment the same region over several recordings over days and match up neurons across days?

*Reviewer #2:*

The manuscript by Zhou and colleagues presents a computational method for extracting calcium signals from neurons in images that are "corrupted" by high background, with a specific interest in microendoscopic recordings. It builds on previous work from Pnevmatikakis and colleagues, differing primarily in the statistical model used to describe the background. The authors present several examples using both simulations and real experimental data to demonstrate the characteristics of their new method. In comparison with their previous method and a PCA/ICA method, the authors show examples where the new method outperforms the previous one.

The manuscript has many strengths, including the application to several different in vivo data sets and the realistic-looking simulated (with available ground truth) data sets. The model also seems thoughtfully designed, and considerable effort was made to ensure that it will be practical at least for data sets of the size of typical endoscopic recordings. The figures are mostly clear except as noted below.

Overall, this may be a worthy addition to the armamentum of methods; however, some aspects of the analysis are difficult to judge, and my overall assessment is that its importance is not yet clear. My major questions/concerns are:

1) First, this reviewer doesn't fully understand why/how the background/foreground separation works. Loosely, the model is of the form `Y ≈ b + a*c`, where `b` is the background and `a` and `c` are the spatial/temporal components of the neurons. In previous work they argued that a non-negativity constraint on `a` and `c` led to substantial improvement over unconstrained methods and effective separation of the neurons from a (small) background. However, in this case `b` is of quite large magnitude, and one might expect this to effectively "relax" the constraint on `a` and `c`: the model could fit a smaller value for `b` in the vicinity of a neuron, allowing `a` and `c` to take on larger positive values and thus "cushioning" them against the impact of the nonnegativity constraint. In a sense, one might imagine that the fully-converged result would not be terribly different from a (sparse, local) singular value decomposition, which is known to not properly segment cells particularly in the presence of temporal correlations among nearby/overlapping cells.

Nevertheless, the temporal traces exhibit the characteristics expected from a "meaningful" nonnegative factorization. So, the question is, why? One possible answer is that the authors are not iterating their model to convergence, and that an intermediate initialization via a "meaningful" nonnegative step still has substantial influence over the final form of the solution. If this is the case, the authors should also present what happens when they iterate their model to full convergence (e.g. square root of the machine precision) and discuss the heuristics they use to choose when to terminate the iteration early.

2) For the real datasets they initialize the neuronal factorization using a high-pass spatial filtering of the raw image, then reintroduce the full image and fit the complete model. But in the end, we're mostly interested in the neurons, and we may not really care that much about an accurate model of the background. How does the quality of the neuronal reconstruction compare if you just use CNMF on the spatially-filtered (and 0-truncated) image? If it performs similarly, it's not entirely clear that this more complex model represents much of an advance.

It's possible that there's a bit of data in the manuscript on this point (Figure 4, black dots), but to this reviewer it's not clearly described, and this is shown only for simulated data. If you don't know the ground truth, would you care about the difference between 0.99 similarity and 0.999 similarity?

3) The authors describe some manual interventions in the methods, but there does not appear to be much in the way of description of exactly how these interventions were leveraged. It would be helpful to readers to understand what the "raw" output of the unsupervised algorithm really looks like, how much manual investment (time, numbers of each type of decision, etc.) is necessary, and how much the manual intervention improves the result.

Related to these points (particularly the first two), how does CNMF-E fare against CNMF when applied to two-photon data? i.e., when the additional flexibility of the model is perhaps not essential, does this extra flexibility degrade the performance in some fashion?

*Reviewer #3:*

This paper introduces the CNMF-E algorithm for extraction of neurons from 1-photon calcium imaging data.

1) First and foremost, CNMF-E has quickly become widely-adopted by scientists who are collecting micro-endoscopic video data. Clearly this paper is very high impact and deserves to be published in a prominent journal. I congratulate the authors for an important piece of work, which will surely have a sustained and important impact on the field.

2) I'm concerned about the semi-manual "interventions" described in subsection “Interventions”. It is mentioned in subsection “in vivomicroendoscopic imaging and data analysis” that manual interventions were applied in the data analysis shown in this paper. This makes the comparisons shown in this paper unfair to competitors that are fully automated (since of course any method can be improved using manual intervention). Can the authors show results of CNMF-E without manual intervention?

3) The CNMF-E algorithm is just a slight tweak on CNMF, which is itself a pretty standard matrix factorization. This lack of statistical innovation is probably OK, though, since *eLife* is not looking to publish innovative statistical methodology, but rather an algorithm that works well and has very high impact.

4) I feel very strongly that the exposition of the CNMF-E algorithm in this paper could stand to be improved substantially. It is extremely hard (or, really, impossible) to make sense of the details of the proposed algorithm as presented in this paper! While the big picture of the algorithm is clear from a quick glance at Equation 8, the details are not clear from the many pages of text that follow. When I read a statistical- or computational-focused paper, I wonder "would a statistically/computationally-literate reader of this paper be able to re-implement this algorithm, from scratch, based on the description of this paper?" The answer to that question is definitely "no". More details about this are provided below.

5) It is really wonderful that the authors have easy-to-use code available for the CNMF-E algorithm! However, it is also very important that they post scripts online recreating the figures shown in this paper, for both the simulated and the real data. For instance, I'd like to see a script that one can simply call into matlab that will read in the mouse dorsal striatum data and output exactly the results shown in Figure 5. (A similar request applies to the other figures). This is particularly important in light of the "semi-manual interventions" mentioned in my point 2 above. Furthermore, the actual data used to make the figures (e.g. the calcium recordings for the mouse dorsal striatum area that were used to make Figure 5) should be made available.

Without the availability of the data and a script to get the results shown in the paper, I think it will be very hard for others to apply CNMF-E and get comparable results, given the many tuning parameters and semi-manual interventions involved. It will also be very hard for others to objectively evaluate how well CNMF-E works, and how sensitive the results in this paper are to tuning parameter selection, etc.!

6) I think that most of the benefit of CNMF-E over CNMF comes from the careful initialization of the algorithm described in subsection “Initialization of model variables”, rather than from the details of the optimization problem Equation 8. What happens if one passes in the initialization described in subsection “Initialization of model variables” to the CNMF algorithm or to other competitors in the literature?

[Editors' note: further revisions were requested prior to acceptance, as described below.]

Thank you for resubmitting your work entitled "Efficient and accurate extraction of in vivo calcium signals from microendoscopic video data" for further consideration at *eLife*. Your revised article has been favorably evaluated by David Van Essen (Senior Editor and Reviewing Editor), and the original three reviewers.

The manuscript has been improved but there are two small remaining technical points that need to be addressed before acceptance, as outlined below. Both should be easy to address.

1) In subsection “Fitting the CNMF-E model”, the optimization problem (P-All) needs to list s and B^f as optimization variables (i.e. they should appear under 'minimize'). While the reviewer agrees with the authors that "s_i is completely determined by c_i and G^{(i)}, and B^f is not optimized explicitly", nonetheless s_i and B^f are optimization variables. This is actually just a fundamental math issue that has nothing to do with the specifics of the paper at hand!

To understand this point, consider the optimization problem "minimize_{x,y} f(x,y) s.t. x=y". It would NOT be correct to instead write this as "minimize_x f(x,y) s.t. x=y", EVEN THOUGH (to quote the authors), "x is completely determined by y". (Why is "minimize_x f(x,y) s.t. x=y" incorrect? Well, unless y is listed as an optimization variable, then it is treated as a constant in the optimization problem; therefore, the solution to that problem is trivially just x=y, with minimum value attained at f(y,y).)

Thus, (P-All) is simply incorrect as written. The fix is simple: put "s_i" and "B^f" under the "minimize".

2) In a couple of places (e.g. subsections “Fitting the CNMF-E model”and” Ranking seed pixels”) it is mentioned that (P-All) is not jointly convex in the optimization variables. Unfortunately, though this statement is true, it is misleading: when we say that a problem is not "jointly convex", we are implying that it is convex in the individual optimization variables. By contrast, (P-All) isn't even convex in the individual optimization variables. This is because details aren't provided of what (for instance) "is sparse" means. If that means that the l1 norm is penalized, then yes, it is convex in the individual optimization variables. But if it means some other notion of sparsity, e.g. that the l0 norm is penalized, then (P-All) isn't convex in the individual optimization variables.

Therefore, the reviewer feels that stating that (P-All) is "not jointly convex" or "jointly non-convex" is misleading and should be removed from the text where it appears. The authors should instead just say "non-convex" without the word "jointly".

---

## [Author Response]

Thank you for your letter and for the thoughtful and constructive comments, which helped us greatly. We believe the revision is a substantial improvement over the original submission.

Essential revisions:1) It is important to discuss carefully to what degree this new algorithm's success reflects more of an accident of initialization than good statistical design. Given that (a) SVD will always produce a lower reconstruction error than NMF for the same number of components, and that (b) having a high background might allow the neural components to "insulate" themselves from the constraints of nonnegativity by offsetting them from zero, it seems possible that the model also admits solutions that are better from the standpoint of the loss function yet worse from the standpoint of biological plausibility. As a consequence, it may only be the initialization causing the algorithm to land in a local (but far from global) minimum that is leading to the kinds of plausible solutions exhibited in the manuscript; the risk is that a more exhaustive optimization algorithm, applied to the same data and model, would prefer solutions that lack this plausibility.

Thanks for these thoughtful points.

Regarding the importance of initialization, we note first that the initialization results are typically not perfect; running the iterative optimization proposed here does lead to quantitatively better solutions, as shown in simulated data in Figure 4 and the new Figure 5. In real data, we often find that the initialization picks out high-SNR neurons but misses low-SNR neurons that can only be detected after further iterations. Second, the results of Figure 4 further show that if we apply a perfect initialization (i.e., initialize with the ground truth) and then apply a statistical model that fits the data poorly (e.g., CNMF with a rank-1 background model) then we obtain significantly worse results than CNMF-E initialized with no prior knowledge of the ground truth. We have clarified both of these important points in the revised text (see Results section). Regarding the reviewer’s interesting point about the SVD: we agree that SVD minimizes a certain approximation norm: svds(Y,k) finds the closest rank-k matrix to Y in terms of the mean-square reconstruction error. However, the resulting singular vectors will typically be very poor estimators of the neuronal components, for reasons that are well-understood: unlike real neural components, the singular vectors will be orthogonal, not sparse, not non-negative, not spatially localized, etc. In short, the SVD is solving the wrong optimization problem, because while it is minimizing a reasonable cost function it is optimizing over the wrong constraint space. We don’t want to search over all possible low-rank matrices to approximate the data matrix Y; instead, to obtain good estimates we need to impose constraints both on the single-neuron components (e.g. sparsity and nonnegativity of neural activity, sparsity and locality of neuronal shapes, etc.) and on the background (which is spatially smooth, as encoded by the novel “ring” background model introduced here). We have tried to impose these constraints in a tractable manner in this work.

So, to address this question of artificially increasing the size of the background: yes, by doing this one could reduce the mean-squared reconstruction error but at the cost of forcing the inferred neural activity to be highly non-sparse and the background to be highly non-smooth in the spatial domain. In other words, this solution would achieve a good objective value (like the SVD solution) but would grossly violate the constraints we have placed on the problem, and thus this solution is not favored by our optimizer – as desired.

Regarding the point about local vs global optima – it is theoretically possible there is a solution that satisfies the constraints and gives better MSE but is less neurally plausible. Due to the non-convexity of the objective function completely exhaustive search seems intractable. But our experience as described in this manuscript (and the experience of our users) has been that solutions that find a good MSE under the objective function are “plausible.” The new Figure 4 provides additional support for this claim; see further discussion below.

2) The exposition of the CNMF-E algorithm in this paper could stand to be improved substantially. It is extremely hard (or, really, impossible) to make sense of the details of the proposed algorithm as presented in this paper! While the big picture of the algorithm is clear from a quick glance at Equation 8, the details are not clear from the many pages of text that follow.

Thanks for this feedback. We have edited the text significantly to clarify a number of details that were not sufficiently clear in the original manuscript. See detailed changes discussed further below.

3) There are concerns about how well the technique would work if the true firing correlations increase. Can the authors do a simulation where they increase pair-wise correlations between the deltaF traces systematically and see at what point segmentation breaks down or cross-talk removal artificially lowers the correlations.

Great suggestion. See the new Figure 5 and the new text paragraph in subsection “initialization of model variables” describing highly spatially- or temporally-correlated data.

4) Please provide more practical advice about how to implement the software. This method is very computationally intensive and some direction needs to be given on how to run the software to allow a large number of movies to be analyzed in a reasonable amount of time. There are no benchmarks given (from my reading) on how long the analysis takes per minute of recording and how this can be optimized.

Good suggestion. We added more practical advice right after the summary of the CNMF-E pipeline (subsection “Initialization of model variables”). We also improved our CNMF-E implementation to support large scale data analysis and to process multiple videos together. Details are included in the revised manuscript (subsection “Ranking seed pixels”). We also provided more timing information. Regarding optimization of timing, we believe further significant speedups could be achieved with more parallelization, implementation in low-level languages, etc., and plan to pursue these directions in the future, but we chose to focus this manuscript on the model, estimation, and demonstrations on real and simulated data, rather than these lower-level timing optimizations.

We are also maintaining wiki pages for CNMF-E (https://github.com/zhoupc/CNMF_E/wiki) for continuously providing practical advice about how to use the software.

5) Please comment on success of being able to segment the same region over several recordings over days and match up neurons across days?

We think CNMF-E should make this task significantly easier (because the output SNR is much improved compared to PCA/ICA), but we have not pursued a quantitative analysis of this point. A recent paper (Sheintuch et al., 2017) addressed exactly this issue and showed that more neurons could be reliably tracked using CNMF-E vs PCA/ICA; we have added a reference to this work.

6) The authors should also present what happens when they iterate their model to full convergence (e.g., square root of the machine precision), and discuss the heuristics they use to choose when to terminate the iteration early.

We added a residual sum of squares (RSS) vs iteration plot in Figure 4. We note in the revised text that we can use the RSS as a convergence criterion, but actually computing the RSS is relatively slow; instead we monitor the estimated neural components A and C, e.g., the number of detected components.

7) How does the quality of the neuronal reconstruction compare if you just use CNMF on the spatially-filtered (and 0-truncated) image? If it performs similarly, it's not entirely clear that this more complex model represents much of an advance.

Yes, this occurred to us as well. We tried this, and it doesn’t work well, because spatial filtering distorts the neural shapes and also removes the useful nonnegativity constraint from the optimization. We have added a note to the text making this point clear.

8) It is mentioned in subsection “in vivo microendoscopic imaging and data analysis” that manual interventions were applied in the data analysis shown in this paper. This makes the comparisons shown in this paper unfair to competitors that are fully automated (since of course any method can be improved using manual intervention). Can the authors show results of CNMF-E without manual intervention? Also, how much manual investment (time, numbers of each type of decision, etc.) is necessary, and how much the manual intervention improves the result.

Fair points. We should clarify one point: it is not quite true that any method can be trivially improved using manual intervention. Specifically, PCA/ICA outputs a collection of spatial & temporal filters, after the user inputs the dataset and the number of principal components and the number of independent components. Users can then easily manually delete components; however, there is not an obvious way to merge or split neurons (e.g., after splitting a neural component it is not clear how to assign temporal traces to the resulting split components). More importantly, there is no mechanism for refining the analysis results after manual interventions (for example, to account for a gap left after deleting some bad components that were partially explaining some of the observed signal). In CNMF-E, on the other hand, we can run additional iterative updates of the estimated components following manual interventions, which in practice can lead to significantly improved results. This point has been clarified in the revised text.

That said, to respond to the main point, we emphasize that the manual intervention in CNMF-E is an optional choice for correcting results using human knowledge, instead of being required by processing data. For the simulated data and three out of the four real datasets in the revised manuscript (subsection “Initialization of model variables”), we now apply no manual interventions at all: the algorithm was run in fully automatic mode. For the more challenging hippocampal dataset, we show the results before and after manual intervention in Video S10.

Of course, fully automated algorithms are preferable where possible. We believe that most of the interventions discussed here (e.g., detection of “bad” components) can be cast as standard classification or computer vision tasks and can in the future be fully automated with sufficient training data; this is the topic of ongoing work. Our open-source pipeline makes it easy to swap in more automated intervention steps when and if these become available and reliable in the future. Again, these points have been clarified in the revised text.

9) How does CNMF-E fare against CNMF when applied to two-photon data? i.e., when the additional flexibility of the model is perhaps not essential, does this extra flexibility degrade the performance in some fashion?

The CNMF-E model also works for 2p data in which the background has simple spatiotemporal structure. This is hinted at in Figure 2, where the CNMF-E model works well for both a small and large number of background sources. Several labs already use CNMF-E for processing their 2p data e.g. obtained via GRIN lenses; we hope to present these results in future work but decided against including these results here to avoid lengthening an already-long paper. That said, for “vanilla” 2p data where the rank-1 background model suffices, the basic CNMF model is preferred in practice, because the resulting inference is faster (though not significantly more accurate) than in the CNMF-E model. In the toolbox we have developed users can easily choose different initialization options and background models depending on the type of data under analysis.

10) It is important to make the data and the scripts available for recreating the figures shown in this paper, for both the simulated and the real data. Otherwise it will be very hard for others to apply CNMF-E and get comparable results, given the many tuning parameters and semi-manual interventions involved.

Agreed. We now provide all code and scripts for generating the figures and the videos (https://github.com/zhoupc/*eLife*_submission)

11) What happens if one passes in the initialization described in subsection “Initialization of model variables” to the CNMF algorithm or to other competitors in the literature?

This is an important point that we emphasize more clearly in the revised text (subsection “CNMF-E accurately initializes single-neuronal spatial and temporal components”). In Figure 4 we initialize CNMF with the ground truth and find that it performs poorly, due to the poor fit of its background model here. (We obtained similar results with real instead of simulated data; in real micro-endoscopic data, performing CNMF iterations significantly reduces the SNR of the estimated traces, due to background contamination.)

Reviewer #1:

This is an excellent paper showing how the CNMF technique has been adapted for use with 1-P calcium imaging in endoscopic data. The paper is very clearly written, and the simulations backed up by the in-vivo imaging data clearly demonstrates that this technique is by far superior to any other currently being used, including ICA/PCA or other variations. In fact, it will be essential to use this technique to obtain the highest quality data as background fluctuations and cross-talk between neurons can severely impact the data recorded.

Thanks very much for your kind comments.

These are my concerns:1) I have some concerns about how well the technique would work if the true firing correlations increase. Can the authors do a simulation where they increase pair-wise correlations between the deltaF traces systematically and see at what point segmentation breaks down or cross-talk removal artificially lowers the correlations.

Please see our responses to the Essential revision question (Bhatia et al., 2015).

2 I felt that the authors could provide more practical advice about how to implement the software. This method is very computationally intensive, and some direction needs to be given on how to run the software to allow a large number of movies to be analyzed in a reasonable amount of time. There are no benchmarks given (from my reading) on how long the analysis takes per minute of recording and how this can be optimized.

Please see our responses to the Essential revision question (4).

3) Can the authors comment on success of being able to segment the same region over several recordings over days and match up neurons across days?

Please see our responses to the Essential revision question (Cameron et al., 2016).

Reviewer #2:

The manuscript by Zhou and colleagues presents a computational method for extracting calcium signals from neurons in images that are "corrupted" by high background, with a specific interest in microendoscopic recordings. It builds on previous work from Pnevmatikakis and colleagues, differing primarily in the statistical model used to describe the background. The authors present several examples using both simulations and real experimental data to demonstrate the characteristics of their new method. In comparison with their previous method and a PCA/ICA method, the authors show examples where the new method outperforms the previous one.The manuscript has many strengths, including the application to several differentin vivo data sets and the realistic-looking simulated (with available ground truth) data sets. The model also seems thoughtfully designed, and considerable effort was made to ensure that it will be practical at least for data sets of the size of typical endoscopic recordings. The figures are mostly clear except as noted below.Overall, this may be a worthy addition to the armamentum of methods; however, some aspects of the analysis are difficult to judge, and my overall assessment is that its importance is not yet clear. My major questions/concerns are:1) First, this reviewer doesn't fully understand why/how the background/foreground separation works. Loosely, the model is of the form `Y ≈ b + a*c`, where `b` is the background and `a` and `c` are the spatial/temporal components of the neurons. In previous work they argued that a non-negativity constraint on `a` and `c` led to substantial improvement over unconstrained methods and effective separation of the neurons from a (small) background. However, in this case `b` is of quite large magnitude, and one might expect this to effectively "relax" the constraint on `a` and `c`: the model could fit a smaller value for `b` in the vicinity of a neuron, allowing `a` and `c` to take on larger positive values and thus "cushioning" them against the impact of the nonnegativity constraint. In a sense, one might imagine that the fully-converged result would not be terribly different from a (sparse, local) singular value decomposition, which is known to not properly segment cells particularly in the presence of temporal correlations among nearby/overlapping cells.Nevertheless, the temporal traces exhibit the characteristics expected from a "meaningful" nonnegative factorization. So, the question is, why? One possible answer is that the authors are not iterating their model to convergence, and that an intermediate initialization via a "meaningful" nonnegative step still has substantial influence over the final form of the solution. If this is the case, the authors should also present what happens when they iterate their model to full convergence (e.g., square root of the machine precision), and discuss the heuristics they use to choose when to terminate the iteration early.

Thanks for these very thoughtful comments.

Re: the `Y ≈ b + a*c`model – yes, but additionally recall that the background term b here is constrained to have a strong degree of spatial smoothness, so the model can’t simply add a “hole” into the new time-varying background term b to somehow help a*c avoid the nonnegativity constraint. In addition, as noted above, the model would also have to violate the sparsity constraint on c to avoid the nonnegativity constraint. We have also significantly revised the “Algorithms for solving problem (P-T)” subsection in the Materials and methods section to clarify this point.

Re: iterating to convergence – our results are not dependent on early stopping. See the new panel B in Figure 4, and also our responses to Essential revision question (Carvalho Poyraz et al., 2016).

2) For the real datasets they initialize the neuronal factorization using a high-pass spatial filtering of the raw image, then reintroduce the full image and fit the complete model. But in the end, we're mostly interested in the neurons, and we may not really care that much about an accurate model of the background. How does the quality of the neuronal reconstruction compare if you just use CNMF on the spatially-filtered (and 0-truncated) image? If it performs similarly, it's not entirely clear that this more complex model represents much of an advance.

See our response to the Essential revision question (Cichocki and Phan, 2009).

To expand on this slightly: we agree that many users may not care much about the background signals, just as many users of extracellular voltage recordings only care about extracted spikes, not local field potentials. But our goal for CNMF-E is to demix and assign all the signals in the data correctly, with minimal spatial or temporal distortion. In our opinion it is likely that the background signal (like the local field potential) carries a lot of useful information which has to date been largely ignored, and we want to avoid corrupting or discarding this information. Without accurate background-subtraction, the extracted temporal components will often be highly contaminated, as we have emphasized in this manuscript (subsection “Model and model fitting”).

It's possible that there's a bit of data in the manuscript on this point (Figure 4, black dots), but to this reviewer it's not clearly described, and this is shown only for simulated data. If you don't know the ground truth, would you care about the difference between 0.99 similarity and 0.999 similarity?

Fair question. In addition to our response above (emphasizing the importance of a full separation of the movie data into properly-defined and interpretable spatial, temporal, and background components), our response to Essential Revision question (Apthorpe et al., 2016) touches on this issue (on fitting a complete model compared to just using the initializations directly). See also the new Figure 5, showing a more direct example of significant improvements of running the full CNMF-E pipeline (not just the initialization).

3) The authors describe some manual interventions in the methods, but there does not appear to be much in the way of description of exactly how these interventions were leveraged. It would be helpful to readers to understand what the "raw" output of the unsupervised algorithm really looks like, how much manual investment (time, numbers of each type of decision, etc.) is necessary, and how much the manual intervention improves the result.

Please see our respondents to the Essential revision question (8).

Related to these points (particularly the first two), how does CNMF-E fare against CNMF when applied to two-photon data? i.e., when the additional flexibility of the model is perhaps not essential, does this extra flexibility degrade the performance in some fashion?

Please see our respondents to the Essential revision question (9).

Reviewer #3:

This paper introduces the CNMF-E algorithm for extraction of neurons from 1-photon calcium imaging data.1) First and foremost, CNMF-E has quickly become widely-adopted by scientists who are collecting micro-endoscopic video data. Clearly this paper is very high impact and deserves to be published in a prominent journal. I congratulate the authors for an important piece of work, which will surely have a sustained and important impact on the field.

Thanks for this kind comment.

2) I'm concerned about the semi-manual "interventions" described in subsection “Interventions”. It is mentioned in subsection “in vivo microendoscopic imaging and data analysis” that manual interventions were applied in the data analysis shown in this paper. This makes the comparisons shown in this paper unfair to competitors that are fully automated (since of course any method can be improved using manual intervention). Can the authors show results of CNMF-E without manual intervention?

Please see our responses to the Essential revision question (8).

3) The CNMF-E algorithm is just a slight tweak on CNMF, which is itself a pretty standard matrix factorization. This lack of statistical innovation is probably OK, though, since eLife is not looking to publish innovative statistical methodology, but rather an algorithm that works well and has very high impact.

We would argue against the comment that there is no statistical innovation here. Much careful, effective, cutting-edge statistical treatment of neural data might be dismissed as "slight tweaks," in this sense. But statistical methods cannot be judged solely from an abstract perspective, and as soon as methods are brought to bear on complicated problems, the "tweaks" constitute a major portion of the research effort. It is a serious statistical challenge to cleanly demix data in which the background artifacts are an order of magnitude larger than the desired single-neuronal signals – and as we have argued here, both PCA/ICA and vanilla CNMF do not do an adequate job of removing this background contamination. (To put it another way, we experimented with many “tweaks” of CNMF that did not work well; this was a highly non-trivial development process.) But that said, we agree that in the end the main differences between CNMF and CNMF-E are a different background model and a different initialization procedure; these two innovations led to an algorithm that works well and that we hope will have a significant positive impact on the field.

4) I feel very strongly that the exposition of the CNMF-E algorithm in this paper could stand to be improved substantially. It is extremely hard (or, really, impossible) to make sense of the details of the proposed algorithm as presented in this paper! While the big picture of the algorithm is clear from a quick glance at Equation 8, the details are not clear from the many pages of text that follow. When I read a statistical- or computational-focused paper, I wonder "would a statistically/computationally-literate reader of this paper be able to re-implement this algorithm, from scratch, based on the description of this paper?" The answer to that question is definitely "no". More details about this are provided below.

Please see our responses to the Essential revision question (Barbera et al., 2016).

5) It is really wonderful that the authors have easy-to-use code available for the CNMF-E algorithm! However, it is also very important that they post scripts online recreating the figures shown in this paper, for both the simulated and the real data. For instance, I'd like to see a script that one can simply call into matlab that will read in the mouse dorsal striatum data and output exactly the results shown in Figure 5. (A similar request applies to the other figures). This is particularly important in light of the "semi-manual interventions" mentioned in my point 2 above. Furthermore, the actual data used to make the figures (e.g. the calcium recordings for the mouse dorsal striatum area that were used to make Figure 5) should be made available.Without the availability of the data and a script to get the results shown in the paper, I think it will be very hard for others to apply CNMF-E and get comparable results, given the many tuning parameters and semi-manual interventions involved. It will also be very hard for others to objectively evaluate how well CNMF-E works, and how sensitive the results in this paper are to tuning parameter selection, etc.!

Please see our responses to the Essential revision question (10).

6) I think that most of the benefit of CNMF-E over CNMF comes from the careful initialization of the algorithm described in subsection “Initialization of model variables”, rather than from the details of the optimization problem Equation 8. What happens if one passes in the initialization described in subsection “Initialization of model variables” to the CNMF algorithm or to other competitors in the literature?

Please see our responses to the Essential revision question (Apthorpe et al., 2016) for explaining the limitations of the initialization step.

To our knowledge, the background model in CNMF-E is the first one that accurately models the background components in this type of data; this new background model (along with our new initialization approach) is the main innovation of our work.

As for the suggestion of passing the initialization to other competitors, we did so using simulated data, where we passed in the ground truth spatial and temporal components to the vanilla CNMF algorithm (Figure 4). The results are much worse than CNMF-E. This is a clear evidence that the low rank NMF background model used in vanilla CNMF is not enough for modeling the background components in microendoscopic data. We would expect similar results with related methods that use a similar low-rank matrix factorization model (e.g. Diego-Andilla and Hamprecht, 2013, or Maruyama et al., 2014). We also tried the similar model in Suite2p (Pachitariu et.al., 2016), which projects the background onto a prespecified spatial basis, and again found that this led to worse results in recovering the background signal in the datasets examined here.

PCA/ICA does not offer an easy method for incorporating warm initialization starts, as discussed further above.

[Editors' note: further revisions were requested prior to acceptance, as described below.]

1) In subsection “Fitting the CNMF-E model”, the optimization problem (P-All) needs to list s and B^f as optimization variables (i.e. they should appear under 'minimize'). While the reviewer agrees with the authors that "s_i is completely determined by c_i and G^{(i)}, and B^f is not optimized explicitly", nonetheless s_i and B^f are optimization variables. This is actually just a fundamental math issue that has nothing to do with the specifics of the paper at hand!To understand this point, consider the optimization problem "minimize_{x,y} f(x,y) s.t. x=y". It would NOT be correct to instead write this as "minimize_x f(x,y) s.t. x=y", EVEN THOUGH (to quote the authors), "x is completely determined by y". (Why is "minimize_x f(x,y) s.t. x=y" incorrect? Well, unless y is listed as an optimization variable, then it is treated as a constant in the optimization problem; therefore, the solution to that problem is trivially just x=y, with minimum value attained at f(y,y).)Thus, (P-All) is simply incorrect as written. The fix is simple: put "s_i" and "B^f" under the "minimize".

Great suggestion. We made three changes:

a) in (P-All), we put 'S' and 'B^f' under the 'minimize'

b) in (P-T), we put 'S' under the minimize

c) in (P-B), we put 'B^f' under the minimize.

2) In a couple of places (e.g. subsections “Fitting the CNMF-E model”and “Ranking seed pixels”) it is mentioned that (P-All) is not jointly convex in the optimization variables. Unfortunately, though this statement is true, it is misleading: when we say that a problem is not "jointly convex", we are implying that it is convex in the individual optimization variables. By contrast, (P-All) isn't even convex in the individual optimization variables. This is because details aren't provided of what (for instance) "is sparse" means. If that means that the l1 norm is penalized, then yes, it is convex in the individual optimization variables. But if it means some other notion of sparsity, e.g. that the l0 norm is penalized, then (P-All) isn't convex in the individual optimization variables.Therefore, the reviewer feels that stating that (P-All) is "not jointly convex" or "jointly non-convex" is misleading and should be removed from the text where it appears. The authors should instead just say "non-convex" without the word "jointly".

Great suggestion. We removed the word ‘jointly’ in all places.